# SHORT-CONTEXT DOMINANCE: HOW MUCH LOCAL CONTEXT NATURAL LANGUAGE ACTUALLY NEEDS?

## ABSTRACT

We investigate the short-context dominance hypothesis: that for most sequences, a small local prefix suffices to predict their next tokens. Using large language models as statistical oracles, we measure the minimum context length (MCL) needed to reproduce accurate full-context predictions across datasets with sequences of varying lengths. For sequences with 1–7k tokens from long-context documents, we consistently find that 75–80% require only the last 96 tokens at most. Given the dominance of short-context tokens, we then ask whether it is possible to detect challenging long-context sequences for which a short local prefix does not suffice for prediction. We introduce a practical proxy to MCL, called Distributionally Aware MCL (DaMCL), that does not require knowledge of the actual next-token and is compatible with sampling strategies beyond greedy decoding. Our experiments validate that simple thresholding of the metric defining DaMCL achieves high performance in detecting long vs. short context sequences. Finally, to counter the bias that short-context dominance induces in LLM output distributions, we develop an intuitive decoding algorithm that leverages our detector to identify and boost tokens that are long-range-relevant. Across Q&A tasks and model architectures, we confirm that mitigating the bias improves performance.

## 1 INTRODUCTION

When prompted to continue a piece of text, how much preceding context do humans rely on? Do they focus on recent words and local coherence, or plan with a broader, narrative-wide perspective? For example, when writing a story, do they recall events from earlier chapters or rely mostly on recent developments? While difficult to study these questions rigorously in humans, large language models (LLMs) offer a tractable experimental analogue. This leads us to ask: *How far back must a model look to predict the next token accurately?* Although modern transformer-based LLMs can attend to thousands of tokens ([Beltagy et al., 2020](); [Dai et al., 2019]()), it remains unclear and unquantified how often that capacity is used at inference time, and how often predictions depend on distant information or primarily on local spans.

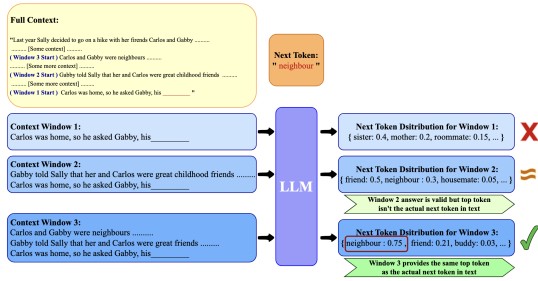

Figure 1: **Short-context dominance hypothesis.** In this example, with context 1 the model fails to predict the ground-truth "neighbour," Context 2 produces a semantically valid alternative "friend," and when using Context 3, the model correctly predicts "neighbour." Our work **(1)** systematically validates the hypothesis, **(2)** develops methods to detect when longer context is truly needed, and **(3)** leverages these insights to improve language model sampling by correcting short-context bias.

We posit the *short-context dominance hypothesis*: for the majority of natural-language sequences, the information required to accurately predict a valid next token is contained within a *short*, *local* prefix of length $k \ll n$, where $n$ is the full-context length of the sequence. Confirming this hypothesis could: (1) reveal how much model capacity is truly needed for next-token prediction of a given sequence, (2) inform the design of better performing sampling methods, and, eventually, (3) identify opportunities for architectural or training modifications that leverage locality for computational efficiency.

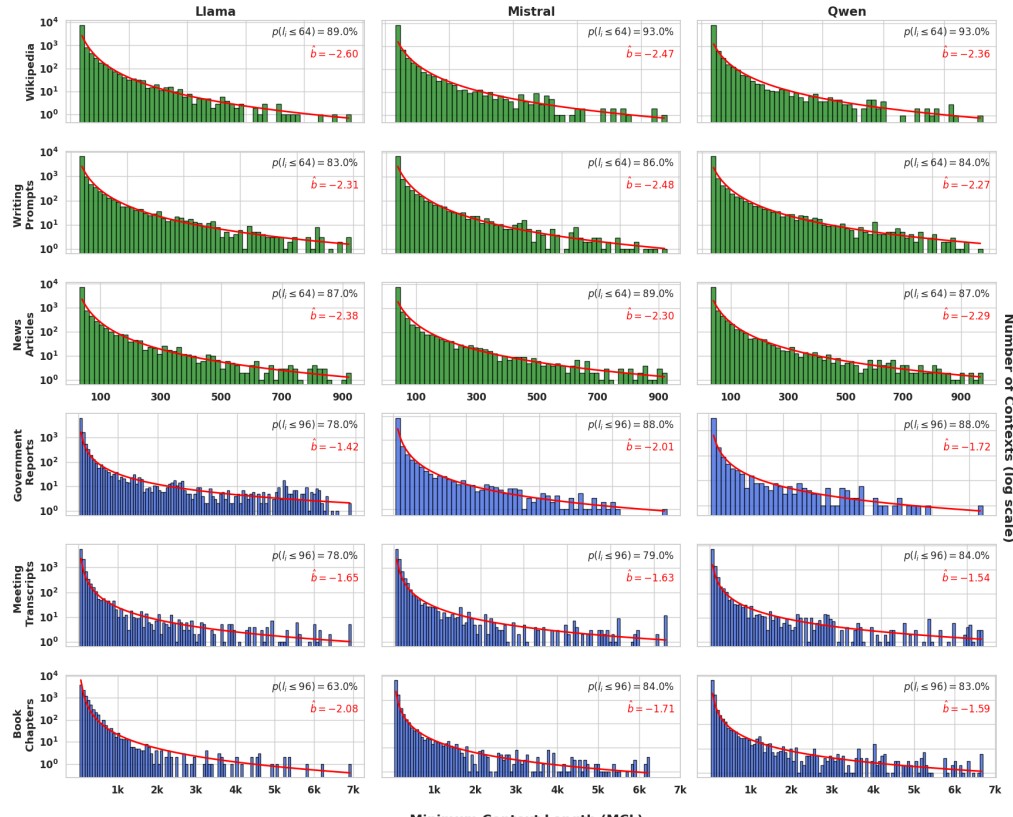

Figure 2: **Distribution of MCL:** Minimum context window needed to confidently predict the next token across sampled sequences, six datasets, and three LLMs. $\hat{b}$ denotes the slope of the log-log fit. Blue/green distinguishes between long ($\geq$ 6k tokens) and short ($\geq$ 1k tokens) documents, respectively.

**Measuring natural-language local context dependency.** To test our hypothesis, we introduce **Minimal Context Length (MCL)**, which quantifies how much local context suffices for a language model, used as oracle, to confidently and correctly predict the ground-truth next token of a sequence. We sample sequences of varying lengths from various sources and filter for cases where an LLM correctly predicts the actual next token in the corpus, aiming to mitigate confounding effects of LLM limitations. To quantify MCL, we iteratively increase the prefix length of each sequence until the model confidently outputs the ground-truth next token. We find consistently that 75-80% of sequences with lengths 100-7k tokens rely on at most 32-96 last tokens, supporting our hypothesis. See Fig. 2.

**Practical long-context detection.** We develop a practical variant of MCL, the **Distributionally Aware Minimum Context Length (DaMCL)**, that does *not* require ground-truth knowledge and is compatible with sampling strategies beyond greedy decoding. We validate that DaMCL remains consistent with the short-context dominance hypothesis. Importantly, we show that it enables accurate classification of sequences as *short-context* (where a 32-token prefix suffices) or *long-context* (requiring longer context). Since detection operates without ground-truth information, it is practical for inference-time applications.

**Post-hoc short-context dominance correction.** With a long-context detector at hand, we finally test a hypothesis that short-context dominance induces a corresponding bias in LLMs themselves: Since sequences predominantly require only short context, models are implicitly trained on distributions heavily skewed toward local dependencies, biasing them toward common completions and filler words predictable from short local context, hurting prediction for long-context sequences (Sharma et al., 2023; Malkin et al., 2022; Duh et al., 2024). We validate such a short-context bias by applying TaBoo (**Ta**rgeted **Boo**sting ), an intuitive decoding algorithm that counteracts the skew: for sequences that we detect as being long-context, TaBoo identifies and boosts tokens that are long-context-relevant.

On Q&A datasets with inherent long-context dependencies, TaBoo consistently outperforms vanilla nucleus sampling and competitive logit-adjustment methods across model architectures.

## 2 RELATED WORK

We summarize the most closely related works below and defer detailed discussion to Appx. B.

**Long context utilization analysis.** Prior studies demonstrate that LMs often underutilize available context (Khandelwal et al., 2018; Sun et al., 2021; Liu et al., 2023). These findings have inspired algorithmic interventions at both system and data levels (Zhang et al., 2024; Borgeaud et al., 2022; Izacard et al., 2022; Chen et al., 2025; Chuang et al., 2025). A more closely related recent study by Fang et al. (2025) provides methodology for detecting tokens with long context relevance and emphasizing them upon evaluation and fine-tuning. Our work differs by providing systematic quantification of minimal context requirements from the perspective of natural language properties themselves, introducing practical detection methods that operate without ground-truth tokens (a key distinction to Fang et al. (2025)), and demonstrating targeted inference-time interventions.

**Contrastive decoding.** Several inference-time methods address hallucination in long-context generation through contrastive approaches (Li et al., 2023; Zhao et al., 2024; Liu et al., 2021). Building on (Li et al., 2016; Brown et al., 2020), Malkin et al. (2022); Duh et al. (2024) reweight distributions by contrasting long vs. short contexts. CAD (Duh et al., 2024) is most closely related to our TaBoo algorithm, but differs fundamentally in both motivation and implementation. While CAD uniformly adjusts output probabilities for all sequences and tokens through contrastive considerations, our algorithm stems from the short-context dominance hypothesis and applies targeted adjustments to long-context sequences and their long-range-relevant tokens. van der Poel et al. (2022) apply entropy-based selection for contrastive decoding, but their method differs again in both theoretical motivation and technical implementation. Beyond distinctions specific to TaBoo, we systematically study minimal context length requirements in natural language and our long-context detection methods could apply beyond inference-time correction.

**n-gram models.** Recent work has revisited n-gram models as complementary to neural language models (Li et al., 2022; Liu et al., 2025; Nguyen, 2024). The success of such inherently short-range models (even Liu et al. (2025)'s Infini-gram typically captures at most 32-token dependencies) provides indirect evidence toward short-context dominance in natural language. Our systematic quantification of minimal context requirements offers a principled explanation of this, supporting the broader thesis that majority of language understanding tasks require primarily local information.

## 3 LEAST CONTEXT FOR PREDICTION

In this section, we answer the following question: *For a given randomly sampled context and next-token, what is the minimum sub-context needed to predict the actual next token correctly?*

### 3.1 MINIMAL CONTEXT LENGTH

We isolate sequences $\mathbf{s}$ for which the LLM, using greedy decoding, **correctly** and **confidently** predicts the actual next token $t$ in the corpus. By focusing on these high-quality predictions, we approximate using the LLM as a statistical oracle to study context dependency. We define MCL as follows.

**Definition 1.** *The **Minimal Context Length** (MCL) of sequence $\mathbf{s}$ given its next token $t$ is the length $\ell$ of the shortest prefix $\mathbf{s}_{[-\ell:]}$ so that the model output given the prefix is correct and confident. Formally,*

$$\mathsf{MCL}\,(\mathbf{s}|t) := \arg\min_{l \in |\mathbf{s}|} \left\{ l \;\middle|\; \mathsf{Top}_1(\mathbf{s}_{[-l:]}) = t, \;\; \Delta\mathsf{Conf}(\mathbf{s}_{[-l:]}) \geq \delta \right\} \tag{1}$$

Here, $\mathbf{s}_{[-\ell:]}$ denotes the prefix of the last $\ell$ tokens of sequence $\mathbf{s}$ of length $|\mathbf{s}|$; $\mathsf{Top}_1(\cdot)$ returns the token with highest model output probability given the input sequence; and $\Delta\mathsf{Conf}(\cdot)$ returns the gap in probability between the top-ranked and second-best token, which must exceed confidence threshold $\delta \in [0, 1]$. For concreteness in our experiments, we set $\delta = 0.2$ (see Appx. C.1 for ablation). By definition, a larger MCL implies that the model requires information from earlier in the context to predict the next token correctly, while smaller MCL indicates greater reliance on local context.

## 3.2 EXPERIMENTAL SETUP

**Datasets.** We experiment on the following natural-language documents. Short documents: Reddit *Writing Prompts* (Fan et al., 2018), CNN/DailyMail *News Articles* (Hermann et al., 2015; Nallapati et al., 2016) and *Wikipedia* articles from WikiText-103 (Merity et al., 2016). Long documents: U.S. *Government Reports* from GovReport (Huang et al., 2021), *Meeting Transcripts* from QMSum Zhong et al. (2021) and story *Book Chapters* BookSum (Kryściński et al., 2022). These appear in the LongBench and LongEval (Bai et al., 2024; Krishna et al., 2023) benchmarks and are believed to have inherent long-context qualities. We further experiment with documents of different languages using (Schwenk et al., 2019) and datasets specialized for math and medical domains (Appx. C.2).

**LLM Oracles.** We evaluate three open-weight models: LLaMA-3-8B (Grattafiori et al., 2024), Mistral-7B-Instruct (v0.2) (Jiang et al., 2023) and Qwen2-7B (Yang et al., 2024). We assume that these models are sufficiently capable to exhibit reliable performance on next-token prediction and question answering tasks. Recall also that we isolate sequences that the models confidently predict the next token. All experiments are performed on a V100 Nvidia GPU with 32GB of memory.

**Choice of sequences.** We form sequences $s$ by parsing documents from the datasets. For short documents we sample 100 unique documents of length at least 1k and only keep their first 1k tokens. For long documents we sample documents of lengths $n \in [6, 7]$k, in order to focus our analysis on long-context inputs. We sample 100 sequences $s$ and their ground-truth next-token $t$ from each document. We filter for sequences with correct and confident predictions. We also make sure to avoid biases toward either shorter or longer seqeunces. Concretely, for our 1k token length windows ($0 - 1$k for regular documents and $6 - 7$k for long context data), we ensure to sample the same number of sequences from sizes $[32, 100], [100, 200], \cdots, [900, 1000]$. Overall, this yields 10k unique sequences (100 sequences × 100 documents) and their respective next tokens for each dataset.

**MCL Algorithm.** To determine the MCL as per Defn. 1, we evaluate a model's predictions using increasing prefix sizes $l \in \{32, 48, 64, \ldots, |s|\}$, starting from 32 tokens and incrementing by 16. For the longer documents, we start from 32 and increment by 64. Starting from 32 tokens is motivated by prior work suggesting this length captures local context beyond classical n-gram statistics (Malkin et al., 2022; Liu et al., 2025; Fang et al., 2025). For each window size, we evaluate the model's output distribution and stop once it confidently predicts the ground-truth next token. In practice, we provide the full input to preserve positional encoding and simulate truncated contexts via attention masking.

## 3.3 RESULTS AND DISCUSSION

Fig. 2 shows that the distribution of MCL $(s|t)$ as defined in Equation 1 is highly skewed (note the histogram y-axis in log scale), indicating that the model requires only the last $32 - 96$ tokens for the majority of contexts ($\sim 80$–90%) to confidently predict the next token (MCL $(s|t) \leq 32$ or 96). This observation formally confirms that **for majority of queries, the LLM only needs highly localized information from the context to correctly and confidently predict the token**.

To quantify short-context reliance at finer granularity, we examine the power-law exponent $\hat{b}$ by fitting $y = a \cdot x^{-b}$ in log-log space. For shorter documents, we observe values of $\hat{b} \in [-2.5, -2]$, and for longer documents, $\hat{b}$ falls in the range $[-2, -1.5]$. Both ranges indicate strong power-law decay, demonstrating short-context sufficiency even in inherently long-context datasets such as *Government Reports*, *Meeting Transcripts*, and *Book Chapters*. We further validate these findings across linguistic and domain variations through experiments detailed in Appx. C.2. Table 3 demonstrates that this pattern holds consistently: **Short-context dominance persists across our experimental setups**.

Validating the short-context dominance hypothesis carries implications for both pretraining and evaluation: Since most predictions require only highly localized information, then standard training objectives and perplexity-based evaluations are inherently skewed toward short-range dependencies, potentially obscuring true progress on long-context reasoning. This helps explain prior findings that only small fraction of tokens benefit from contexts larger than 2K (Sun et al., 2021). It also reinforces recent critiques (Fang et al., 2025) that token-level perplexity is an insufficient metric for long-context evaluation—even on datasets with genuine long context dependencies, such as *Government Reports*.

## 4 DISTRIBUTIONAL AWARENESS

MCL evaluates whether the ground-truth next token from the dataset can be predicted with a shorter prefix. Intuitively, natural language often permits multiple valid next tokens, and models may

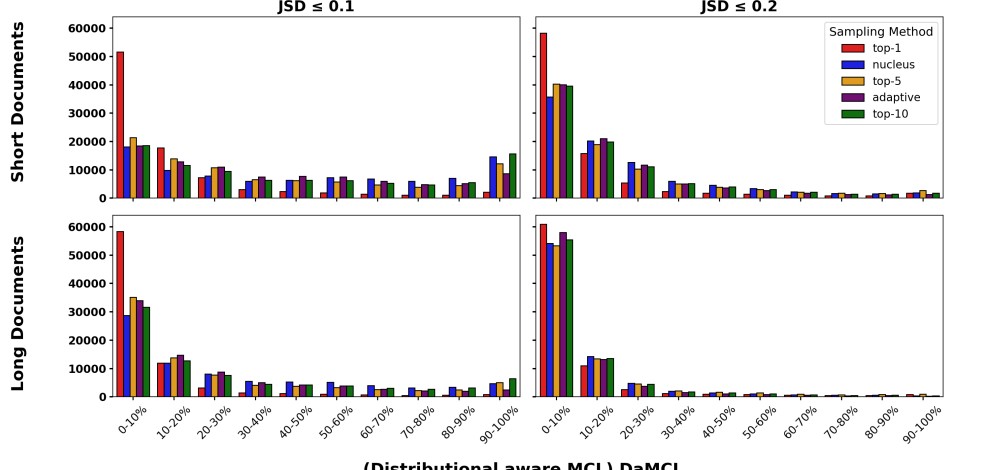

Figure 3: **Distribution of DaMCL:** DaMCL measurements for various sampling strategies and relative thresholds. Results are shown separately for short- (top row) and long- (bottom row) documents to highlight potential differences in behavior. While the overall trend resembles the heavy-tailed decaying pattern observed in standard MCL (see Fig. 2), the choice of threshold influences the outcome. Each subplot reflects results aggregated over all model–dataset combinations, as only minor deviations were observed across different configurations under identical hyperparameters.

assign high probability to plausible alternatives that differ from the ground-truth token in the dataset. Definition 1 is also limited to greedy decoding, while popular natural language generation methods often rely on sampling strategies that draw from multiple probable tokens (Holtzman et al., 2020; Basu et al., 2021; Zhu et al., 2024). These considerations (see also Appx. D.1) motivate us to introduce here a more flexible MCL formulation.

## 4.1 DISTRIBUTION AWARE MCL

Consider decoding strategy $\phi$ that modifies the model's raw probability distribution. For example, nucleus sampling (Holtzman et al., 2020) selects a subset of tokens whose cumulative probability mass reaches threshold $p$ (e.g. $p = 0.9$), then produces a renormalized distribution $\mathbf{p}_\phi(\mathbf{s})$ with support on this subset and zero probability to tokens otherwise. More generally, our framework accommodates any decoding strategy $\phi$ that produces a probability distribution $\mathbf{p}_\phi(\mathbf{s})$. We introduce DaMCL to find the smallest prefix $\mathbf{s}_{[-\ell:]}$ such that $\mathbf{p}_\phi(\mathbf{s}_{[-\ell:]})$ is sufficiently similar to $\mathbf{p}_\phi(\mathbf{s})$.

**Definition 2.** *The **Distribution-aware Minimal Context Length** (DaMCL) of a sequence $\mathbf{s}$, given decoding strategy $\phi$, a probability-similarity metric $\mathcal{M}$, and a similarity threshold $\epsilon$, is defined as the length of the shortest prefix for which the decoding-based next-token distribution given the prefix $\mathbf{p}_\phi(\mathbf{s}_{[-\ell:]})$ lies within an $\epsilon$-neighborhood of the full-context distribution $\mathbf{p}_\phi(\mathbf{s})$. Formally, we define:*
$$\mathrm{DaMCL}_\phi^{\mathcal{M}}(\mathbf{s}, \epsilon) := \arg\min_{l \in |\mathbf{s}|} \left\{ l \mid \mathcal{M}(\mathbf{p}_\phi(\mathbf{s}_{[-l:]}) \, ; \, \mathbf{p}_\phi(\mathbf{s})) \le \epsilon \right\}.$$

While this definition accommodates any distance metric $\mathcal{M}$, we specifically use the Jensen-Shannon Distance (JSD) throughout our experiments. For distributions $\mathbf{p}_1, \mathbf{p}_2$ in the $|\mathcal{V}|$-dimensional simplex, JSD is defined as $\mathrm{JSD}(\mathbf{p}_1 \, ; \, \mathbf{p}_2) := \sqrt{\frac{1}{2}\mathrm{KL}(\mathbf{p}_1\|\mathbf{q}) + \frac{1}{2}\mathrm{KL}(\mathbf{p}_2\|\mathbf{q})}$, where $\mathbf{q} := \frac{1}{2}(\mathbf{p}_1 + \mathbf{p}_2)$, and $\mathrm{KL}(\mathbf{p}_1\|\mathbf{p}_2) := \sum_{t \in \mathcal{V}}[\mathbf{p}_1]_t \log \frac{[\mathbf{p}_1]_t}{[\mathbf{p}_2]_t}$ is the Kullback-Leibler divergence. We choose JSD because it is a proper distance metric satisfying the triangle inequality and is widely employed in knowledge distillation (Gu et al., 2024) and language model evaluation (Ji et al., 2023a). The above definition of DaMCL (1) relies on measuring the difference between distributions as opposed to a single token, and, (2) does not require ground-truth information regarding the actual next token of sequence $\mathbf{s}$.

### 4.2 Experimental Setup

**DaMCL evaluation.** We evaluate DaMCL over several decoding strategies: Top-$K$ sampling ($K$=1 for greedy) with $K \in [10]$ (Radford et al., 2019; Fan et al., 2018), nucleus sampling with $p$=0.9 (Holtzman et al., 2020), and adaptive sampling with $\epsilon$=0.001 (Zhu et al., 2024). We evaluate two threshold values JSD $\leq 0.1$ and JSD $\leq 0.2$ to study the effect of varying similarity criteria.

Lower thresholds correspond to stricter conditions for accepting a local prefix as equivalent to the full context. Complementary to the setup of Sec. 3.3, we use lengths $\ell$ of prefixes based on percentiles of the full sequence (specifically, the last 10% to 100%) rather than fixed-length truncation. The length $|\mathbf{s}|$ of the full sequences is restricted to [6, 7]k for long docs and [200, 1000] otherwise.

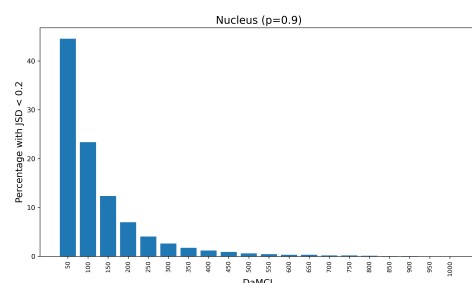

Figure 4: DaMCL distribution for nucleus sampling with fixed sub-context increments. Still heavily biased towards short context.

### 4.3 Results and Discussion

To begin with, Fig. 3 for JSD $\leq 0.2$ reveals a similar bias toward short context as with MCL. Yet, note that the drop in DaMCL is not as dramatic as observed when using MCL suggesting that, from a distributional standpoint, larger portions of context are required to ensure similarity compared to the ground-truth-token-based MCL.

When enforcing stricter similarity requirements (JSD $\leq 0.1$) we find the distribution becomes flatter and even experiences a U-shaped structure. This suggests that a larger subset of tokens either resolve with short contexts or require nearly the full input to reach distributional convergence. We further analyze this in Appx. D.3 and discover that this bimodality is mainly attributed to the shorter sequences. Indeed, we find that the long documents, for which all sequences have more than 6k tokens, do not experience the bimodality to the same degree.

Further, while all decoding strategies show a similar long-tailed behavior for JSD $\leq 0.2$, for the more strict conditions their DaMCL flatness differs. While top-1 sampling continues to exhibit standard decay across settings, broader sampling methods such as nucleus, top-5, top-10, and adaptive sampling increasingly lead to flatter distribution. As these methods spread probability mass across a wider support set, the resulting distributions become smoother and more diffuse, making convergence under tight JSD thresholds more difficult.

For completeness, we also provide results for fixed sub-context lengths in Fig. 4, where we use fixed increments of 50 (for efficiency). These results suggest that short context dominance still remains even when using distribution shift as the metric for detecting minimal context length, even if the severity is somewhat reduced compared to greedy decoding from MCL. Further results are in Appx. D.3.

## 5 Long-Context Sequence Detection

We have seen that for the majority of sequences a valid next-token can be generated with access to only a short local prefix. Specifically, in Sec. 4, to avoid the need for knowing the actual next-token of a sequence, we introduced the idea of quantifying whether a prefix is a good proxy of the full-context by evaluating the JSD of the respective model output probabilities. Here, we build on these insights to develop a long-context detector that can distinguish between *short-context sequences*, where a short prefix of fixed length suffices, and *long-context sequences*, which require longer context information.

### 5.1 Distributional-aware long-context sequence detection

To develop our long-context detector, we first define the following metric.

**Definition 3.** *The **Long-Short Distribution Shift (LSDS)** of sequence $\mathbf{s}$ is the JSD between the next-token distributions obtained with decoding strategy $\phi$ when given a short prefix of length 32 versus the full context. Formally,* $\mathsf{LSDS}(\mathbf{s}) = \mathrm{JSD}(\mathbf{p}_\phi(\mathbf{s}_{[-32:]}), \mathbf{p}_\phi(\mathbf{s}))$.

For concreteness, unless otherwise stated, we fix $\phi$ to nucleus sampling with parameter $p = 0.9$ (Holtzman et al., 2020). (Ablation results on different metrics and $p$ values in Appx E.2.) Finally, we fix the prefix length to 32 based on our findings in Sec. 3 and consistent with Liu et al. (2025).

**Controlled validation.** To demonstrate that LSDS can effectively detect long-context sequences, we conduct a controlled needle-in-a-haystack experiment adapted from Kamradt (2023). We create short and long queries, such that only in the former the answer appears in the final 32 tokens; see Appx. E.1.1 for details and example illustration. Figure 5 shows the resulting LSDS distributions for Mistral-7B-Instruct (v0.2). We observe a clear separation between categories despite no ground-truth next-token information: short-context sequences concentrate at low LSDS values ($\leq 0.4$), while long-context sequences dominate high LSDS values ($\geq 0.7$). Similar patterns hold for LLaMA-3-8B/Qwen2-7B (see Appx. E.1.1).

**Long-context detector.** Based on these findings, we define our long-context detector as a simple thresholding classifier: $\text{LSDS}(\mathbf{s}) \gtrless_{\text{short}}^{\text{long}} \tau$ where $\tau$ is a threshold determined from validation data or prefixed. In the following sections, we analyze the behavior of the detector on natural text.

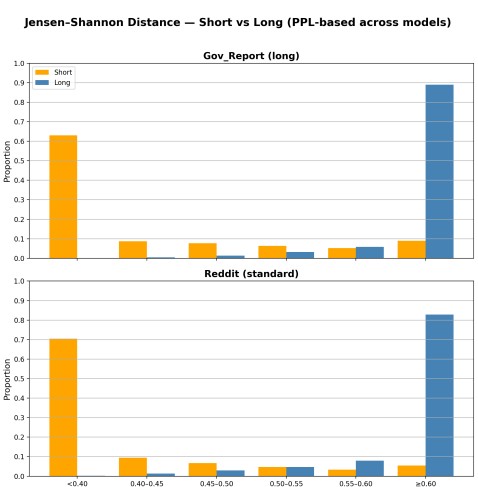

Figure 5: Distribution of LSDS $(\mathbf{s})$ values for short- and long-context sequences for Mistral-7B-Instruct on controlled validation setup.

### 5.2 EVALUATION ON NATURAL TEXT

In natural text, determining the true context dependency requires an oracle with access to the actual next token. We employ two such oracles to establish ground-truth short-context vs long-context labels for sequences. (1) *MCL Oracle:* For $\mathbf{s}$ with next-token $t$, we classify $\mathbf{s}$ as long-context iff $\text{MCL}(\mathbf{s}|t) \geq 32$ (see Defn. 1). (2) *LSD Oracle:* Following Fang et al. (2025), we classify $\mathbf{s}$ as long-context iff $\text{LSD}(\mathbf{s}|t) > 2$ & $\text{LCL}(\mathbf{s}|t) \geq -1$ (details in Appx. E.1.4 ).Both oracles require knowledge of ground-truth next token and thus cannot be deployed at inference time. We show that LSDS-based classification agrees well with these oracles despite not knowing the ground-truth.

**Results.** Figure 6 demonstrates strong agreement between LSDS and the LSD Oracle across *Government Reports* and *Reddit Writing Prompts* datasets. Using $\tau = 0.6$, we find consistently $80 - 90\%$ of oracle-labeled long-context sequences being classified by LSDS as long-context, while fewer than $5 - 10\%$ of short-context sequences are mislabeled. This pattern holds consistently across LLaMA-3-8B/Mistral-7B-Instruct/Qwen-2-7B and both oracles (Appx. E.1), verifying LSDS as a reliable proxy for short vs long context detection.

**Threshold robustness.** Through extensive ablations deferred to Appx. E.1, we find that threshold choices are robust across models and datasets. While a fixed threshold works well generally, task-specific tuning can optimize precision-recall trade-offs.

**Ablation on short-context length.** We test prefixes-lengths $\{8, 16, 32, 64\}$ on Mistral-7B-Instruct-v0.2 (GovReport). Consistent with intuition, too short prefixes $(8, 16)$ lead to larger overlap between short/long distributions, while our chosen values $(32, 64)$ show clearer separation with fewer false positives. We also show good performance particularly with respect to consistency among datasets when adaptively setting the context length to $0.1|\mathbf{s}|$ . See Appx. E.3.

Figure 6: Distribution of LSDS $(\mathbf{s})$ for short- and long-contexts on 2 GovReport (long doc) and Reddit Writing Prompts (short doc), pooled across three models. Illustrates a clear distinction between long and short contexts.

**Computational overhead.** LSDS requires one extra short forward plus JSD calculation beyond normal generation. On Qwen2.5-1.5/7/14B, this adds 35–67 ms across sizes, which is negligible at long contexts, e.g. $\approx 6 - 8\%$ at $|\mathbf{s}| = 6000$. See Appx. E.4.

## 6 LONG-CONTEXT TOKEN BOOSTING

Fewer exposures to long-context sequences may creates a potential bias toward completions that follow from local context, overshadowing long-range dependencies. *Can we identify tokens in the vocabulary that are more relevant to full-context information rather than local context?*

If so, we could favor generating such tokens when we detect a sequence requires long-context reasoning (which, as shown in the previous section, we can reliably identify). Here, we show this is possible and evaluate our method on text-based question answering, a standard benchmark for assessing long-context reasoning capabilities (Liu et al., 2023; Krishna et al., 2023; Beltagy et al., 2020; Bai et al., 2024).

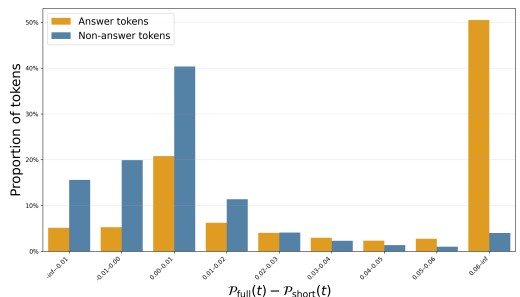

Figure 7: An example illustrating how a shift in the model's next token probabilities when given the long vs short context could help us recognizing the more relevant tokens. Here, boosting the probability of said tokens leads to a more accurate next token distribution.

### 6.1 IDENTIFYING LONG-CONTEXT-RELEVANT TOKENS

First, we show how to identify tokens that are more relevant to full-context information than to local context. Consistent with our theme, the insight is that tokens requiring long-range dependencies should exhibit larger probability increases when given access to full versus short context.

**Definition 4.** *The Long-Short Probability Shift (LSPS) of a vocabulary token $t$ given sequence $\mathbf{s}$ is defined as the change in the assigned probability moving from short to full context under decoding $\phi$:*

$$\mathsf{LSPS}\,(t|\mathbf{s}) = \big[\mathbf{p}_\phi(\mathbf{s})\big]_t - \big[\mathbf{p}_\phi(\mathbf{s}_{[-32:]})\big]_t \;.$$

Here, for probability vector $\mathbf{p}$, $[\mathbf{p}]_t$ denotes its $t$-th entry. As in the previous section, we fix the prefix length to 32 and, unless otherwise stated, for $\phi$, we use nucleus sampling ($p = 0.9$).

Our hypothesis is that tokens genuinely requiring long-context information will show positive LSPS values, as their relevance becomes apparent only with access to the complete context. Conversely, tokens predictable from local context should show small or (even) negative shifts.

**Validation on NarrativeQA.** We validate this hypothesis using *NarrativeQA* (Kočiský et al., 2018). We focus on QA pairs where answers are 1–2 words (typically character names or locations), filtering out cases resolvable without the story context. Given a context with the story, question, and partial answer, we classify the ground-truth next token as an *Answer Token* and all other vocabulary tokens as *Non-Answer Tokens*. Figure 8 shows LSPS distributions for both categories using LLaMA-3-8B.

Figure 8: An example showing how a shift in the long- vs. short-context next-token distribution can signal most long-context-relevant tokens.

Observe the clear separation: Answer tokens exhibit significantly higher LSPS values, confirming that long-context relevant tokens can be identified through their probability shifts. Concretely, using threshold $\epsilon \in [0.05, 0.07]$, we capture over 50% of answer tokens while maintaining low false positive rates ($< 5\%$) on non-answer tokens. Our objective is capturing long-context-relevant tokens while avoiding irrelevant ones, which this approach achieves effectively. In Appx. F, we compare our method to a modified version that uses log-probability ratio (rather than difference) common in prior work (Duh et al., 2024; Malkin et al., 2022; van der Poel et al., 2022; Fang et al., 2025), demonstrating the superior stability and precision of our difference-based approach.

### 6.2 TARGETED LONG-CONTEXT TOKEN BOOSTING

Knowing how to (1) detect long-context sequences using LSDS and (2) identify long-context relevant tokens via LSPS, we now combine these tools to improve Q&A performance. The core insight is to

Table 1: F1, BLEU, and ROUGE-L (Average over all generations and Best-per-example in parentheses). Bold = best Average, Underline = best Best-per-example. We omit results for LLaMA-2-7B on MultifieldQA-en because its context window (4,096 tokens) is insufficient to process the long input passages in this dataset. We include the standard errors (SE) in Appx. F Table 8 for reference.

| Model | Method | NarrativeQA | | | HotpotQA | | | MultifieldQA-en | | |
|---|---|---|---|---|---|---|---|---|---|---|
| | | F1(↑) | BLEU(↑) | ROUGE-L(↑) | F1(↑) | BLEU(↑) | ROUGE-L(↑) | F1(↑) | BLEU(↑) | ROUGE-L(↑) |
| LLaMA-2-7B | Vanilla | 16.1 (36.8) | 2.9 (8.3) | 22.4 (46.2) | 25.2 (51.8) | 7.7 (17.0) | 32.5 (61.0) | NA | NA | NA |
| | CAD | 22.5 (43.3) | 4.3 (9.7) | 30.7 (51.6) | 28.3 (53.1) | 8.5 (16.7) | 34.3 (59.8) | NA | NA | NA |
| | TaBoo | **24.1** (44.9) | **4.9** (10.9) | **31.7** (53.5) | **32.8** (56.4) | **10.3** (18.7) | **39.6** (63.3) | NA | NA | NA |
| LLaMA-3-8B | Vanilla | 24.0 (48.8) | 4.9 (12.1) | 32.7 (57.9) | 29.2 (56.3) | 9.0 (18.8) | 41.6 (68.6) | 19.9 (35.7) | 7.7 (16.6) | 24.0 (41.1) |
| | CAD | **35.4** (55.3) | **7.3** (13.5) | **49.4** (63.5) | 27.7 (46.6) | 8.5 (14.7) | 46.9 (65.3) | 18.8 (33.8) | 6.4 (13.8) | 26.2 (43.1) |
| | TaBoo | 32.0 (53.5) | 7.2 (14.7) | 42.3 (62.8) | **33.1** (55.6) | **10.6** (19.1) | **48.1** (69.6) | **21.9** (35.4) | **9.1** (18.2) | **28.2** (44.0) |
| Mistral-7B-v0.1 | Vanilla | 25.7 (49.7) | 5.1 (12.2) | 33.7 (58.4) | 33.0 (60.1) | 10.1 (19.8) | 43.1 (69.2) | 20.6 (33.6) | 6.9 (14.2) | 26.0 (41.3) |
| | CAD | 34.3 (53.4) | 7.1 (13.6) | 43.1 (62.7) | 35.9 (58.7) | 11.0 (18.8) | 41.6 (64.3) | 18.8 (32.7) | 5.9 (12.8) | 24.9 (41.8) |
| | TaBoo | **35.3** (55.2) | **7.7** (15.0) | **44.4** (64.6) | **37.1** (59.3) | **11.7** (19.7) | **46.3** (67.5) | **23.0** (37.0) | **8.6** (16.3) | **29.5** (45.3) |
| Qwen2-7B | Vanilla | 33.6 (53.1) | 8.1 (14.8) | 42.4 (62.9) | 59.4 (80.5) | 20.1 (29.0) | 62.9 (82.7) | 31.1 (44.9) | 15.1 (25.9) | 41.3 (59.7) |
| | CAD | 36.6 (50.2) | 8.8 (13.6) | 45.3 (59.9) | 59.3 (75.6) | 20.5 (27.4) | 62.2 (78.1) | 30.6 (43.8) | 14.0 (23.5) | 40.0 (55.3) |
| | TaBoo | **38.5** (53.6) | **9.7** (15.4) | **48.2** (63.6) | **63.2** (79.2) | **21.7** (28.4) | **66.6** (81.6) | **32.3** (45.3) | **15.3** (4.7) | **42.3** (58.8) |

promote those identified long-context relevant tokens, downplaying biases from the short context. Simultaneously, this can help downplaying tokens that are assigned high probability as an artifact of noise (Sharma et al., 2023), biases rooted in training data due to potential word/token imbalances (Razeghi et al., 2022; Kassner et al., 2020) and hallucination (Ji et al., 2023b).

**TaBoo (Targetted) Boosting.** Our algorithm TaBoo modifies the probability distribution as per the above intuition. Algorithm 1 details the complete procedure: (1) Detect long-context sequences using LSDS with threshold $\gamma$, (2) Identify long-context relevant tokens where LSPS $\geq \epsilon$, and (3) boost their probabilities by factor $\lambda$ before renormalization and nucleus sampling. See Algorithm 1 in Appx. F.

**Experimental setup.** We evaluate on NarrativeQA (Kočiský et al., 2018), HotpotQA (Yang et al., 2018), and MultiFieldQA (Bai et al., 2024) from the LonBench dataset. Focusing on stories with $\geq$ 1000 tokens to emphasize long-context dependencies, we sample 3,000 examples from NarrativeQA and HotpotQA while using the full 150 examples from MultiFieldQA. We set $\gamma = 0.1225$[1] and $\epsilon = 0.05$. We test on LLaMA-2-7B, LLaMA-3-8B, Mistral-7B, and Qwen-2-7B. For each question-text pair, we generate 5 answers using nucleus sampling and report both average and best-of-5 F1/BLEU/ROUGE scores. We compare our TaBoo approach against two baselines: (1) vanilla nucleus sampling, and (2) Context Aware Decoding (CAD) (Malkin et al., 2022; Duh et al., 2024) with $\alpha = 0.5$, which applies probability adjustments to all generations and their tokens, unlike our targeted boosting of only long-context relevant tokens in detected long-context sequences.

**Results.** Table 1 shows TaBoo consistently outperforms vanilla nucleus sampling across all models and datasets. Compared to CAD, TaBoo achieves superior F1 performance on 11 out of 12 dataset-model combinations, losing only on NarrativeQA with LLaMA-3-8B (although not wrt BLEU score). The improvements generalize across model architectures and scale to higher-performing base models like Qwen2-7B. Best-per-example scores also favor TaBoo, with gains over vanilla ranging from 0.5–8.1 F1 points and generally outperforming CAD as well. Additional results in Appx. F.1, F.4 present ablations on hyperparameter selection ($\gamma, \epsilon, \lambda$). Experiments for short summarization on XSUM (Narayan et al., 2018) are deferred to Tab. 7 in the appendix. Additionally, in Appx. E.4 we show that the computational overhead of LSDS (and thus TaBoo) is minimal for long sequences.

# 7 OUTLOOK

Our work provides a systematic framework for understanding context dependency in language models, with implications for inference efficiency in tasks like QA through principled post-hoc decoding. Specifically, our ability to detect long-context requirements without ground-truth tokens opens opportunities for context-aware generation methods, but also for more targeted evaluation of long-context capabilities and improved training approaches (e.g., following Fang et al. (2025)). Beyond immediate applications, our findings reveal the short-context dominance hypothesis as an inherent property of natural language sequences, with potential broader implications for understanding how language models process and generate text, such as providing alternative motivations for recently proposed logit-adjusted decoding modifications in the hallucination literature Duh et al. (2024).

---

[1]The boosting threshold $\gamma = 0.1225$ is intentionally more liberal than the classification threshold $\tau = 0.6$ used in Sec. 5. While $\tau$ was set conservatively for clear evaluation of long vs. short context, $\gamma$ captures borderline cases where targeted boosting may still be beneficial.

# 8 REPRODUCIBILITY STATEMENT

Following reproducibility requirements, we have ensured to include all detail regarding experimental setup in the text. Regarding the implementation of comparable methods or baseline, we have followed the source reference and common practice to the best of our abilities. Additionally, we have python files and notebooks to recreate versions of our experiments on MCL, long-context detection and the implementation of our TaBoo algorithm.

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

# A  Summary of paper's notations

Table 2 provides a summary of the notation used in this paper.

| Symbol | Description |
|---|---|
| $a$ / $\mathbf{a}$ | Scalar / vector (boldface) |
| $[i]$ | $i$-th entry of vector |
| $[i:j]$ | Index set $\{i, i+1, \ldots, j\}$ |
| $[-l:]$ | Suffix of of length $l$ |
| $\mathcal{V}$ | Vocabulary of tokens, $V = |\mathcal{V}|$ |
| $\mathbf{s} = [t_1, \ldots, t_n]$ | Tokenized document of length $n$ |
| $\mathbf{s}_{[i]}$ | Prefix of $\mathbf{s}$ of length $i$ |
| $\Delta^n$ | Probability simplex in $\mathbb{R}^n$ |
| $\mathsf{Top}_k(\cdot)$ | Indices of $k$ largest entries of a vector; |
| $\pi_\theta(\mathbf{s})$ | LM distribution over $\mathcal{V}$ given context $\mathbf{s}$ |
| $\Delta\mathsf{Conf}(\mathbf{s})$ | Confidence = gap between top-1 and top-2 probabilities |
| $\mathsf{LSDS}(s)$ | Long–short distribution shift (JSD of short vs full context) |
| $\mathsf{LSPS}(t|s)$ | Long–short probability shift for token $t$ in sequence $s$ |

Table 2: Notation used throughout the paper.

# B  Discussion and Related Work

**Pursuit of Long Context:** Capturing dependencies that extend beyond a few tokens has been a long-standing difficulty in language modeling. Early statistical and neural models either truncated context to short n-grams or attempted to maintain memory through recurrence, but both approaches faced limitations with sparsity or vanishing gradients (Chen & Goodman, 1996; Bengio et al., 2003; Hochreiter & Schmidhuber, 1997; Cho et al., 2014). The Transformer architecture (Vaswani et al., 2017) marked a turning point, with self-attention providing a scalable mechanism for integrating information from across the entire input. Contemporary open-access models such as LLaMA 3 (Grattafiori et al., 2024), Mistral (Jiang et al., 2023), Qwen2 (Yang et al., 2024), and Gemma (Team et al., 2024) support context lengths from 8K to 128K tokens, large enough to encode entire novels in a single pass. To make such extensions feasible, architectural innovations like rotary position encodings (RoPE) (Su et al., 2021), attention linear biases (ALiBi) (Press et al., 2021), and position interpolation (Chen et al., 2023) enable models to extrapolate beyond their training horizon, while retrieval-augmented designs (Borgeaud et al., 2022; Izacard et al., 2022; Wang et al., 2023) surface or cache relevant information as an alternative to enlarging the raw attention window. Yet, greater architectural capacity does not imply that models make effective use of long-range context at inference time—the focus of our analysis.

**Context Utilization:** Despite larger context windows, studies show that LMs often do not utilize long-range information. Khandelwal et al. (2018) characterize predictions as "sharp nearby, fuzzy far away," with sensitivity concentrated in the most recent span, while Sun et al. (2021) demonstrate that only a small fraction of tokens benefit from context beyond the first few thousand tokens. Such works emphasize the inherent recency bias in language models. Complementing these results, Liu et al. (2023) document a strong positional bias—models attend to evidence at the edges of long prompts while neglecting the middle, a challenge further analyzed by Zhang et al. (2024), who trace the effect to rotary positional encodings and propose positional modifications to resolve it. To mitigate these limitations, document retrieval approaches have been proposed to surface relevant passages at inference time (Borgeaud et al., 2022; Izacard et al., 2022), while recent work emphasizes the role of training data quality in enabling long-context utilization. In particular, Chen et al. (2025) introduce an attention-based dependency measurement framework (LADM) to identify long documents with strong internal dependencies, showing that selecting such high-quality data for continual pretraining substantially improves long-context performance. Beyond interventions at the system and data level, attribution-based methods such as Chuang et al. (2025) directly test the necessity and sufficiency of context spans, offering a finer-grained perspective on how LLMs actually use long inputs. Similarly, we study the utilization of context from a minimal required sub-context viewpoint, showing that

majority of next token queries utilize very local information. Additionally, we provide methodology to identify the minimal required sub context and while distinguishing between long-short context queries.

**Contrastive Decoding:** Beyond architectural and data-level interventions, several ad hoc inference-time methods aim to improve token generation. Early work encouraged generation that depends more strongly on the given context rather than defaulting to frequent or generic outputs. (Li et al., 2016) propose a mutual-information–based objective for dialogue generation to discourage generic responses, while (Brown et al., 2020) address this issue in multiple-choice QA by normalizing candidate likelihoods against unconditional probabilities, thereby encouraging context-dependent answers. Malkin et al. (2022) introduced coherence boosting, a general inference-time method that reweights the next-token distribution to favor predictions supported by the full long context by explicitly contrasting it against the distribution induced by a shortened context. van der Poel et al. (2022) proposed an entropy-aware decoding strategy for summarization, where the scoring function switches to pointwise mutual information between the source document and the next token when model uncertainty is high, thereby discouraging hallucinations and reducing the tendency to select high-frequency but unsupported tokens. Similar to Malkin et al. (2022) but operating on a logit level, Duh et al. (2024) introduce Context-Aware Decoding (CAD), an inference-time method designed to reduce hallucinations by explicitly amplifying the difference between a model's output probabilities with and without the provided context. Much of this line of research grows out of the broader family of contrastive decoding methods (Li et al., 2023; Zhao et al., 2024; Liu et al., 2021) which are designed for ad hoc modification toe next token distribution to improve language generation. For our case, instead of focusing on performance we first provide an objective study of short context bias of language and use our findings to design intuitive and explainable algorithms to to identify long long contexts, relevant tokens and how to combine this knowledge to help improve generation.

**n-gram and its short context relevance:** Recent work has revisited n-gram models as complements or interpretive tools for neural language models. Li et al. (2022) showed that residual learning with a small n-gram LM can regularize neural text generation and reduce hallucination. More recently, Liu et al. (2025) introduced Infini-gram, which scales n-gram models to trillions of tokens and supports unbounded context length using suffix arrays, enabling both strong next-token prediction and new diagnostic analyses of neural LMs. In parallel, Nguyen (2024) argue for understanding Transformers through the lens of n-grams, showing that simple n-gram rulesets can approximate a majority of model predictions (e.g., covering 68–79% of top-1 predictions across benchmarks). Given that n-grams are inherently short-range models—even Infini-gram typically captures at most 32-token dependencies—the fact that they achieve performance comparable to Transformers on standard text suggests a structural bias in language toward short-context sufficiency, which allows such models to achieve moderate success despite their limited horizon. Our observations regarding the overwhelming prevalence of inherently short contexts in natural language can help explain the reason behind the success of such methods which rely on local information using n-grams for generation.

**Long Context Evaluation:** Much of the evaluation of a model's contextual understanding has focused on tasks such as question answering, retrieval, and needle-in-the-haystack probing, evaluated on datasets such as NarrativeQA (Kočiský et al., 2018), TriviaQA (Joshi et al., 2017), QuALITY (Pang et al., 2022), and LongBench (Bai et al., 2024). While these benchmarks test a model's ability to extract specific information from distant context, they differ from standard language modeling and tend to be highly task-specific. A recent study by Fang et al. (2025) proposes a method for identifying tokens with long-context dependencies and encourages training-time metrics that distinguish such tokens. While their work focuses on a binary classification of long- vs. short-context tokens, we adopt a more fine-grained perspective: treating the language model as a probabilistic oracle and estimating the minimal context required for each next-token prediction in natural text. Additionally, we provide methods for detection of long-context and tokens with long-context relevance without requiring the actual ground truth next token, making our methods applicable during inference.

**Decoding Strategies:** Given our assumption that language models serve as strong proxies for language understanding, it is important to account for the decoding strategy used during inference. A growing body of research has shown that different sampling methods—such as greedy decoding, top-$k$ sampling (Radford et al., 2019; Fan et al., 2018), nucleus ($p$) sampling (Holtzman et al., 2020), and adaptive techniques (Basu et al., 2021; Zhu et al., 2024)—can substantially influence output diversity, factuality, and calibration. Greedy decoding in particular has been shown to produce degenerate or overly deterministic outputs, while adaptive and dynamic approaches aim to adjust

Table 3: Estimated power-law exponents ($\hat{b}$) for MCL distributions on the Wikimedia dataset using Qwen.

|  | English | Arabic | French | German | Chinese | Russian | Korean | Thai |
|---|---|---|---|---|---|---|---|---|
| $\hat{b}$ | $-2.63$ | $-2.41$ | $-2.33$ | $-2.36$ | $-2.35$ | $-2.41$ | $-2.32$ | $-2.30$ |

sampling entropy and generate a high-quality, contextually valid subset of tokens (Holtzman et al., 2020; Zhu et al., 2024; Basu et al., 2021). When treating the language model as a statistical oracle for analyzing context usage, it is essential to consider how decoding strategy influences conclusions about effective context length. This perspective may help improve the practical utility of methods such as Fang et al. (2025), which focus primarily on a single sample from the next-token distribution to classify tokens by their context length requirements. Accordingly, we provide a dedicated analysis of how decoding strategies impact context dependence in next-token prediction and its relation to short-long context detection.

# C  ADDITIONAL RESULTS ON MCL

## C.1  ABLATION RESULTS: MODEL SIZE AND ONFIDENCE THRESHOLD $\delta$

We perform ablation experiments on MCL results regarding model size and the impact of confidence threshold $\delta$ in Fig. 9 and Fig. 10 respectively. Observations further confirm that the short-context dominance hypothesis appears irrespective of experimental setup.

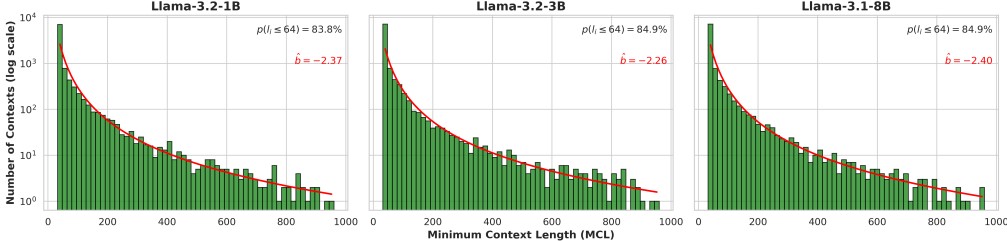

Figure 9: **Impact of model size:** Similar setup to Fig. 2 but Llama models withe different sizes. We can see that the distribution of MCL behavior doesn't change across different models sizes of 1,3 and 7 billion parameter sizes.

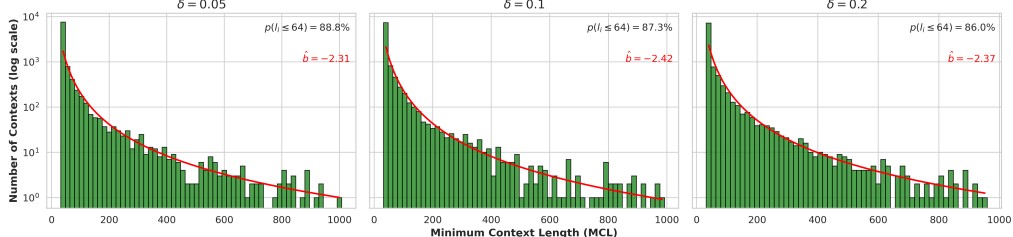

Figure 10: **Impact of confidence threshold** $\delta$**:** Similar setup to Fig. 2 with Llama-3 8B running MCL experiments with different values of $\delta \in \{0.05, 0.1, 0.2\}$. Results suggest that the exponential decay persists irrespective of the confidence threshold.

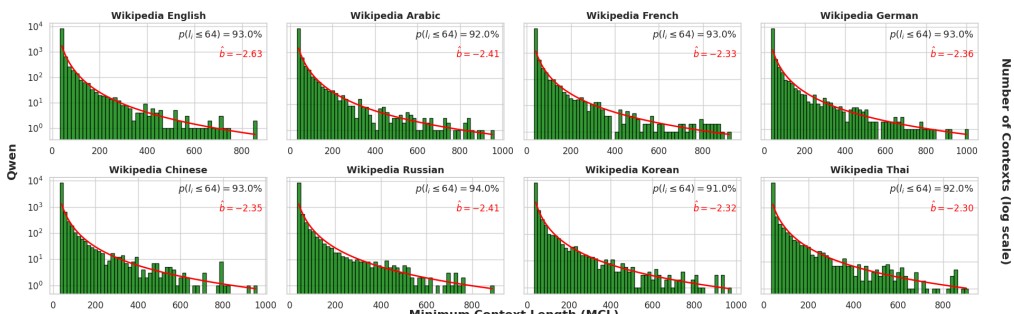

Figure 12: **Distribution of MCL on Different Languages:** Similar setup to Fig. 2 but performed over a set of articles from Wikipedia and their respective translations into different languages. We see that the same trend appears irrespective of target language. This analysis was conducted on Qwen-2.

## C.2 LANGUAGE AND DOMAIN RESULTS

We conduct two sets of experiments to confirm the robustness of the short-context dominance with respect toe different languages and domains. First, we examine the language effect by running inference on Wikipedia article translations in Arabic, Chinese, French, German, Korean, Russian, and Thai, using the Qwen2-7B model (Yang et al., 2024). These texts are sampled from the Wikimedia dataset (Schwenk et al., 2019), and follow the same selection and truncation procedure as our English WikiText documents. Results in Table 3 and Fig. 12 confirm that the reliance on short context is preserved across languages, with similarly highly skewed token count distributions and consistent heavy-tailed fits. Additionally, we perform a small test analyzing the Chain-of-Thought (CoT) generated text by the S1 model (Muennighoff et al., 2025), analyzing the MCL using Qwen-2.5-7B to validate the short-context dominance. Noticeably, looking at the exponential decay in Fig. 11, we see that the CoT contexts exhibit a strong short-context dominance.

Second, to evaluate domain knowledge effects, we test on three specialized datasets: CCDV PubMed Summarization (Cohan et al., 2018) for biomedical abstracts, Open Web Math (Paster et al., 2023)for mathematical content, and LCC_Python (Microsoft, 2024) for code documents, across all 3 models: LLaMA-3-8B (Grattafiori et al., 2024), Mistral-7B-Instruct (v0.1) (Jiang et al., 2023) and Qwen2-7B (Yang et al., 2024). More specifically, the contexts selected from the math and medical datasets are within 1024 tokens, while the context length varies from 6000 to 7000 tokens on the code dataset. As shown in Fig. 13 and Fig. 14, the short-context dominance pattern remains intact across all three domains and models. This suggests models' reliance on local context persist even when dealing with knowledge-intensive content.

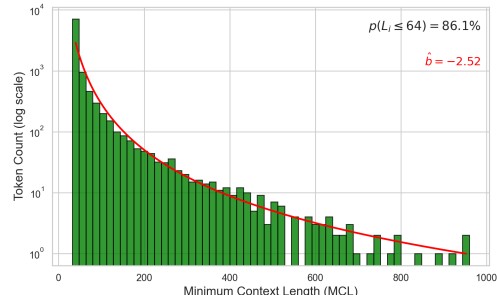

Figure 11: MCL distribution for Qwen2.5 evaluated on context extracted from the generated CoT text generated by the S1 model. We observe that the short context dominance persists and is slightly stronger on this reasoning model generated dataset.

## C.3 MCL RESULTS ON WORD/SUBWORD TOKENS

While we demonstrated the short-context dominance of language in the setting of LLMs, one natural question arises: *Could this phenomenon be an artifact of subword tokenization?* For example, a word like establishments may be tokenized into est, _ablish, and _ments. Intuitively, the model might require more context to correctly predict the first subword compared to the later ones. This raises the question: does the model require less context to guess a non-starting subword (e.g., _ments) than a starting one?

To investigate this, we partition tokens into three inline categories: (1) **full-word tokens**, where a token corresponds to an entire word; (2) **subword-starting tokens**, which occur at the beginning

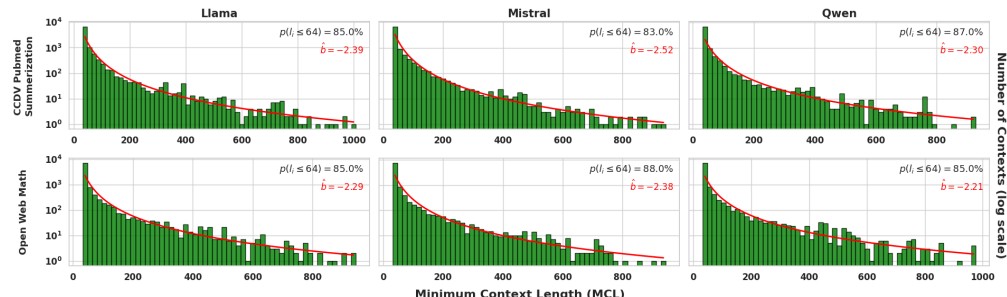

Figure 13: **Distribution of MCL on Math and Medical Datasets:** This figure shows the distribution of MCL values over two specialized domains: medical (CCDV PubMed Summarization) and mathematical (Open Web Math), with a similar setup as Fig. 2. The MCL distribution consistently follows the same trend across both domains and all three models. This suggests that the observed phenomenon is robust to changes in domain knowledge.

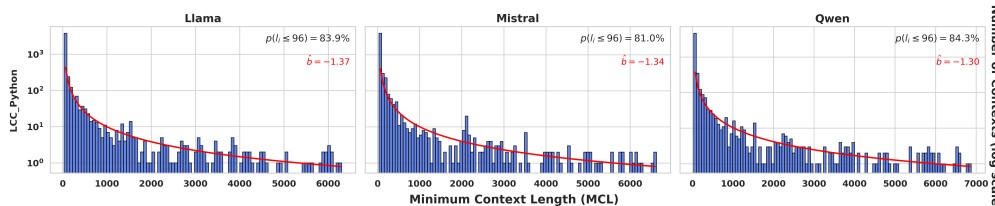

Figure 14: **Distribution of MCL on Code Datasets:** This figure shows the distribution of MCL values over the specialized domain: code (LCC_Python), with a similar setup as Fig. 2. The MCL distribution consistently follows the same trend across all three models.

of a word; and (3) **subword non-starting tokens**, which appear in the middle or end of a word. We then compute MCL separately for these categories, using Mistral-7B-Instruct on a news dataset and Qwen2-7B on writing prompts. As shown in Fig. 16, we observe no meaningful differences in the MCL distributions across the three tokenization categories. If tokenization breaks were driving short-context dominance, we would expect noticeably different decay patterns for full-word versus subword tokens. Instead, the curves remain highly similar.

## C.4 MCL Results on Part-of-Speech

Similar to our analysis on word–versus–subword token behavior, we investigate whether there is any relationship between a token's Part-of-Speech (POS) category and its tendency to exhibit short-context domination. In particular, we ask whether certain types of words—such as nouns or adjectives—are inherently more or less likely to rely on long-range context.

Because POS categories are defined at the word level rather than the token level, we first identify each word in the sequence and assign it to one of four categories: noun, verb, adjective, or adverb. All other words and tokens (e.g., determiners, punctuation, and special tokens) are excluded from our analysis. Each subword token then inherits the POS label of the word it belongs to. We follow a sampling strategy similar to our earlier MCL experiments while keeping the number of tokens per category roughly balanced.

We evaluate two setups: Mistral-7B-Instruct-v0.2 on the CNN News dataset and Qwen2-7B on the Reddit Writing Prompts dataset. Results are presented in Fig. 16. We find that short-context dominance persists across all POS categories. However, compared to nouns and verbs, tokens corresponding to adjectives and adverbs show a slightly greater tendency toward long-context dependence.

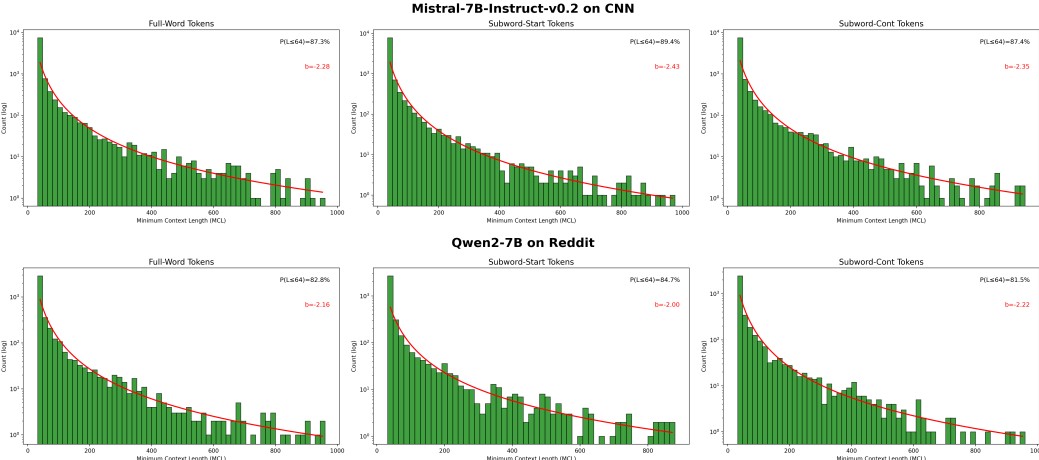

Figure 15: **Distribution of MCL on sub-word tokens:** This figure illustrates the distribution of MCL values for two models and two datasets, separated according to whether the next token is a 1) *full word token*, 2) *sub-word token starting a word* and *sub-word non-starting token* . The experiment illustrates no major difference between the tokens MCL distributions irrespective of its full-word/sub-word.

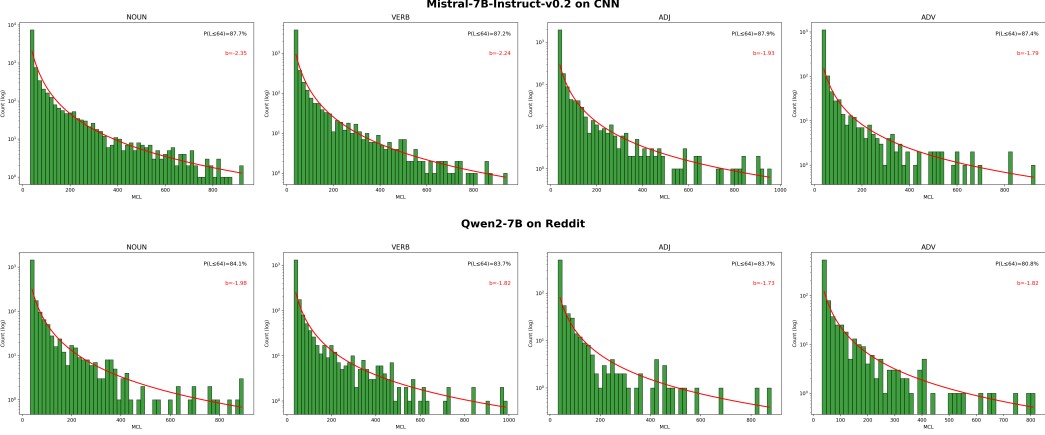

Figure 16: **Distribution of MCL on tokens based on Part-of-speech labeling:** This figure shows the distribution of MCL values for two models and two datasets, separated according to whether the next token belongs to a word in one of the four POS categories: noun, verb, adjective, or adverb. The results reveal that short-context dominance persists across all categories. However, the long-tail behavior suggests that tokens associated with adjectives and adverbs tend to exhibit greater long-context dependence compared to nouns and verbs.

# D DAMCL

## D.1 MOTIVATIONS ON DAMCL

In Sec. 3, we posed the question of determining the minimum subcontext prefix needed to predict the next token in a given dataset. A key limitation of this formulation is that it is constrained by the specific realization of the natural language distribution underlying that dataset.

Put simply, given a context, there are often multiple valid next tokens—valid in terms of the underlying (but unknown) distribution of natural language. While we cannot access this true distribution, we have treated pretrained LLMs as statistical oracles. However, in defining MCL

in Definition 1, we constrain these oracles by evaluating them against only the actual next token from the dataset. Furthermore, we rely solely on greedy decoding, which outputs a single token, thereby underutilizing the model's full predictive distribution as a language oracle. We summarize the issues as follows:

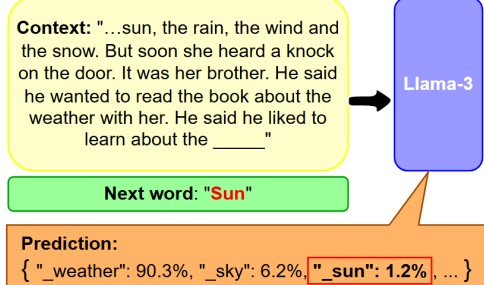

1. Even if the oracle's top-1 prediction does not match the next token in the source text, i.e., $\mathsf{Top}_1(\mathbf{s}_{[-l:]}) \neq t$, this does not invalidate the model's output or imply a lack of contextual understanding. As shown in Fig. 17, the model assigns high probability to several plausible continuations, even if the dataset token is not ranked first. This suggests that relying solely on the dataset token may mislead any context-length detection method.

2. Using the Top-1 token from the sampling distribution is not always a reliable way to evaluate next-token prediction, as greedy decoding often results in low-quality or repetitive outputs (Holtzman et al., 2020). More recent sampling strategies instead aim to identify a set of valid next tokens (Zhu et al., 2024; Zhou et al., 2025), shifting the focus away from single-token probabilities toward broader support coverage.

Figure 17: An example illustrating the limitations of relying on greedy sampling and the actual next token: the Top-1 prediction is a valid response but does not match the ground-truth token. Metrics like MCL may overlook such cases, misidentifying the minimal required context.

These issues motivate the need for a broader definition of MCL—one that 1) **relies on the model's own next-token distribution rather than the actual next token**, and 2) **accounts for the sampling strategy used during inference**. The goal of DaMCL is to mitigate these limitations and offer a more faithful metric for contextual understanding.

### D.2 ADDITIONAL METRICS FOR DaMCL

In addition the main JSD metric used in Sec. 4, we perform the same experiments with a number of other common metrics used to represent similarity between sets and distributions. For the given distributions $\mathbf{p}_1, \mathbf{p}_2 \in \Delta^{\mathcal{V}}$ in the $|\mathcal{V}|$-dimensional simlex we use the following standard distributional similarity metrics:

$$\textbf{Total Variation Distance:} \quad \mathrm{TVD}(\mathbf{p}_1, \mathbf{p}_2) := \frac{1}{2}\|\mathbf{p}_1 - \mathbf{p}_2\|_1 \,,$$

$$\textbf{Kullback-Leibler Divergence:} \quad \mathrm{KL}(\mathbf{p}_1\|\mathbf{p}_2) := \sum_{v \in \mathcal{V}} \mathbf{p}_1(v) \log \frac{\mathbf{p}_1(v)}{\mathbf{p}_2(v)} \,.$$

These metrics are widely employed in applications such as knowledge distillation and in assessing performance degradation of large language models under various conditions (Gu et al., 2024; Ji et al., 2023a; Jia, 2024).

Furthermore, we consider metrics which rely on the inclusions of tokens in the support set rather than the probability distributions. To this end, we define define the Recall, Precision and F1 metric from set $P$ to set $Q$ as:

$$\mathsf{Recall}\,(\mathbf{p}_1 \mid \mathbf{p}_2) := \frac{|\mathbf{p}_1 \cap \mathbf{p}_2|}{|\mathbf{p}_2|} \in [0, 1] \qquad \mathsf{Prec}\,(\mathbf{p}_1 \mid \mathbf{p}_2) := \frac{|\mathbf{p}_1 \cap \mathbf{p}_2|}{|P|} \in [0, 1] \,,$$

$$\mathsf{F1}\,(\mathbf{p}_1 \mid \mathbf{p}_2) := \frac{2 \times \mathsf{Recall}\,(\mathbf{p}_1 \mid \mathbf{p}_2) \times \mathsf{Prec}\,(\mathbf{p}_1 \mid \mathbf{p}_2)}{\mathsf{Recall}\,(\mathbf{p}_1 \mid \mathbf{p}_2) + \mathsf{Prec}\,(\mathbf{p}_1 \mid \mathbf{p}_2)} \in [0, 1] \,.$$

**Recall** measures the proportion of elements in set $\mathbf{p}_2$ that are also present in set $\mathbf{p}_1$, i.e., how much of $\mathbf{p}_2$ is recovered by $\mathbf{p}_1$. **Precision**, on the other hand, quantifies the proportion of elements in

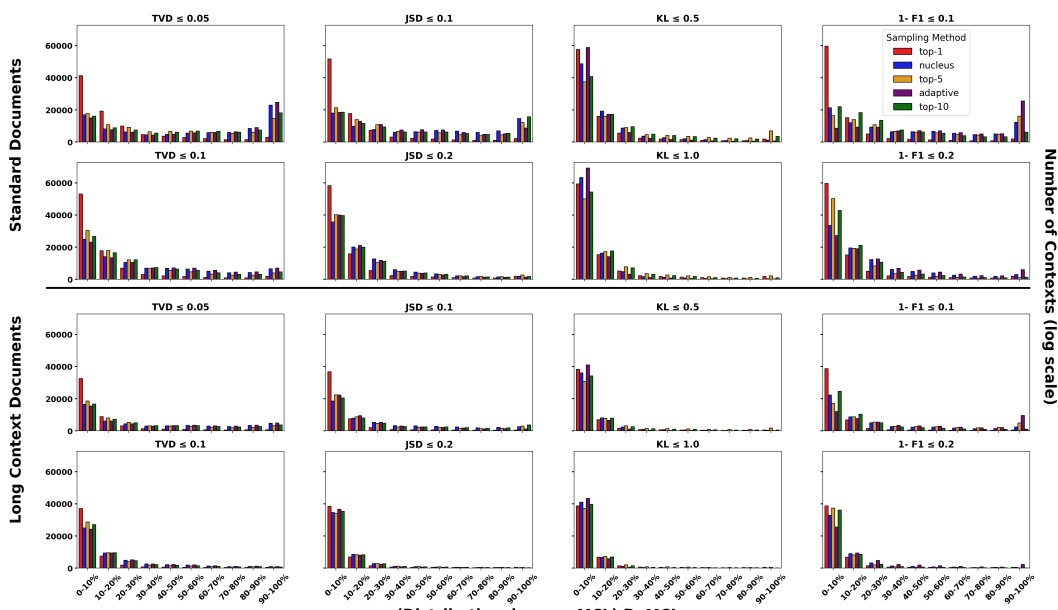

Figure 18: **Distribution of DaMCL:** The distributionally-aware MCL is defined using a range of metrics applied with relative thresholds. Results are presented separately for standard documents (top row) and long-context documents (bottom row) to highlight potential differences in behavior. While the overall trend resembles the exponentially decaying pattern observed in standard MCL (see Fig. 2), the choice of metric and threshold clearly influences the outcome. Each subplot reflects results aggregated over all model–dataset combinations, as only minor deviations were observed across different configurations under identical hyperparameters.

$\mathbf{p}_1$ that are relevant—those that also belong to $\mathbf{p}_2$. A Recall of 1 implies $\mathbf{p}_2 \subseteq \mathbf{p}_1$, meaning all elements of $\mathbf{p}_2$ are captured by $\mathbf{p}_1$. Conversely, a Precision of 1 implies $\mathbf{p}_1 \subseteq \mathbf{p}_2$, indicating that $\mathbf{p}_1$ contains no extraneous elements outside of $\mathbf{p}_1$. The **F1** score is defined as the harmonic mean of Recall and Precision, providing a balanced measure that accounts for both. These definitions are standard for analysis of set similarity and coverage. We specifically calculate $1 - \mathsf{F1}$ in order to match the preference for smaller values, similar to the JSD, TVD and KL metrics.

In comparison to JSD, we observe that both TVD and KL exhibit similar bimodal distribution trends under stricter threshold values. Moreover, neither metric shows substantially different behavior across sampling strategies. In contrast, the $1 - \mathsf{F1}$ metric, while following the same general trend, displays several notable deviations. Specifically, the distributions tend to be more heavily biased toward requiring longer subcontexts to meet threshold requirements. This effect is particularly pronounced in smaller documents when using adaptive sampling, where DaMCL values skew more strongly toward full-context reliance under stricter thresholds.

Our focus on JSD is motivated by several desirable properties that make it especially well-suited for DaMCL. Notably, JSD is a proper distance metric and satisfies the triangle inequality. Furthermore, as a smoothed and symmetric variant of KL divergence, JSD is generally more robust to noise and less sensitive to zero-probability events—properties that are particularly beneficial when working with LLM next-token distributions.

Table 4: **MCL vs. DaMCL results for LLaMA-3.** GovReport is shown at 96 tokens, CNN and Wiki at 64 tokens.

| Dataset | MCL (%) | DaMCL (%) |
|---|---|---|
| GovReport ($\leq 96$) | 75.19 | 36.36 |
| News Articles ($\leq 64$) | 80.52 | 45.43 |
| Wikipedia ($\leq 64$) | 78.15 | 31.65 |

Nonetheless, we acknowledge that further exploration of alternative metrics may reveal additional insights or complementary advantages. We leave this to future work.

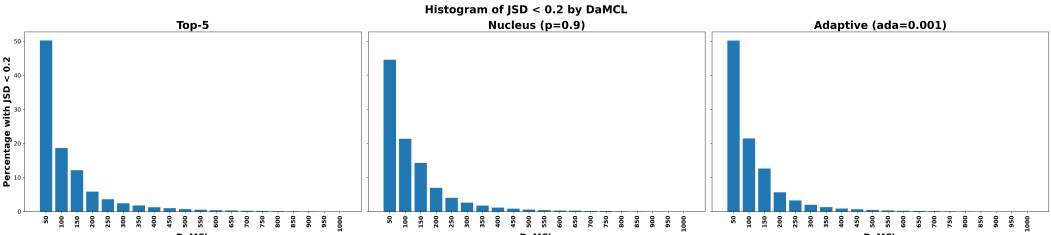

Figure 19: **DaMCL with fixed sub-contexts**: Similar to Fig. 4, but under difference decoding methods.

### D.3    DaMCL with Fixed Sub-context

In order to further analyze the behavioral change of DaMCL compared to MCL, where a larger number of contexts rely on the full length, we perform a number of experiments and analysis using fixed subcontext lengths. First note the results provided in Table 4 which indicate that compared to MCL, a smaller portion of context queries can be resolved with the short sub-contexts.

Looking at Fig. 4 and its more complete counterpart Fig. 19 we still see the bias towards shorter context but less than that of MCL. Once again for ease of computation, we change the sub-context increments to 50 tokens instead of 32. This allows us to analyze the DaMCL values for context of different length separately, allowing us to observe any difference in behavior between longer and shorter sequences.

In Fig. 20, we observe that for longer contexts, the model rarely requires the full context to achieve a close next-token distribution in terms of JSD. In contrast, for shorter contexts ($|\mathbf{s}| \leq 200$), the model more frequently depends on the full context, which may explain the bimodal DaMCL values observed in Fig. 3. For a threshold of $0.2$ (Figure a), only a small fraction of longer contexts lead the model to utilize the entire sequence. Under a stricter threshold of $0.1$ (Figure b), the distribution becomes more uniform and exhibits clearer bimodal characteristics. This suggests that earlier parts of the context can exert a noticeable influence on the next-token distribution when applying tighter similarity thresholds—a potential direction for future investigation.

## E    Long-Context Sequence Detection

### E.1    Additional Setups and Oracles

Here, we provide further experiments with various setups and oracles that complement the results in Section 5. These additional studies reinforce the effectiveness of JSD as a robust detector of long-context dependence.

### E.1.1    Controlled Validation Experiment on LongEval Task

Using the LongEval benchmark (Li* et al., 2023), we construct prompts containing multiple register lines, each with a *line_id* and *<REGISTER_CONTENT>* value, where the model must identify specific content. We create two types of queries with known ground-truth labels: *short-context* queries where the answer (a *line_id*) appears in the last 32 tokens, and *long-context* queries where the answer (a *<REGISTER_CONTENT>*) requires information from the full context. We conduct the experiments on three models: LLaMA-3-8B (Grattafiori et al., 2024), Mistral-7B-Instruct (v0.2) (Jiang et al., 2023), and Qwen2-7B (Yang et al., 2024). As shown in Figure 23, across all models, we observe the same distributional trend, confirming the robustness of the results.

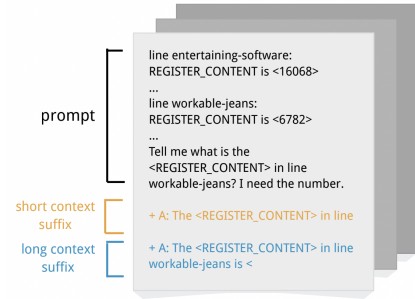

Figure 21: An example of the controlled validation experiment setup on the LongEval benchmark.

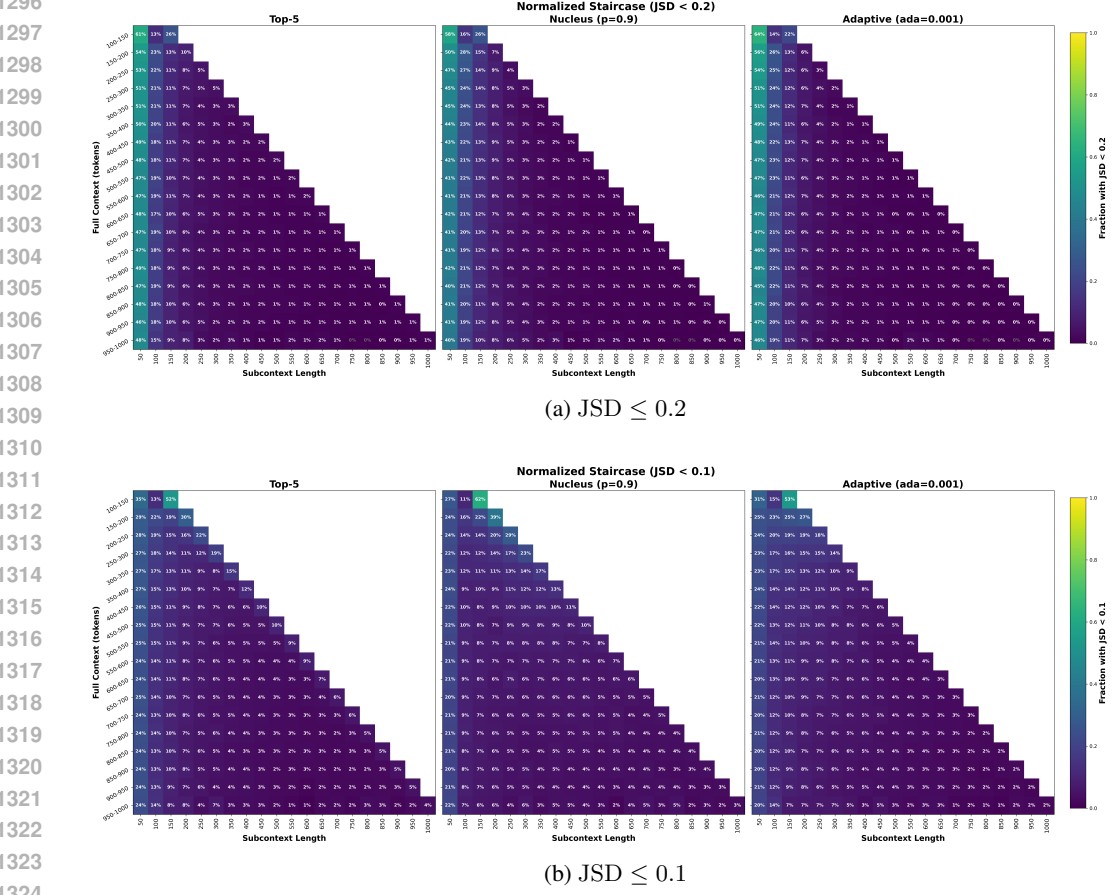

(a) JSD ≤ 0.2

(b) JSD ≤ 0.1

Figure 20: **DaMCL heatmaps with fixed sub-contexts.** Each row represents contexts of a certain size and each column represents sub-contexts tested for different JSD thresholds. Subplots show results for (a) $JSD \leq 0.2$ and (b) $JSD \leq 0.1$.

### E.1.2 NEEDLE-IN-A-HAYSTACK EXPERIMENT ON GENERAL TEXT

We next adapt a needle-in-a-haystack (NIAH) style setup to general text. Following the classic needle-in-a-haystack test by Kamradt (2023), given a natural context, we insert a needle statement with a randomly generated 6-digit number: *"The magic number is xxxxxx"* at a specific position in the text. At the end of the document, we append a query prompt: *"The magic number mentioned in the provided text is "*, expecting the model to output the correct number from the needle. If the needle statement is placed within the final 32 tokens, the answer is recoverable from the local suffix, and we label the case as a short context. Otherwise, when the needle is inserted far away, answering requires long-range recall, and we label it as a long context. We only retain cases where the model outputs the correct number, ensuring the prediction truly relies on the inserted statement rather than hallucination.

Figure 22 presents the JSD distributions under this setup, showing that long-context tokens yield consistently higher JSD values compared to short-context tokens, validating JSD as a detector in this natural-text scenario.

### E.1.3 ADDITIONAL LONG-SHORT CONTEXT DETECTION ON LONGEVAL

In addition to the setup described above, which for the most part follows a similar structure to that of (Fang et al., 2025), we consider a different setup where instead of using the *line_id* as the token representing a short context token, we use *<REGISTER_CONTENT>* in a recent line (less than 32 tokens away). Compared to above, the *long-context* queries remain the same, but for the *short-context*

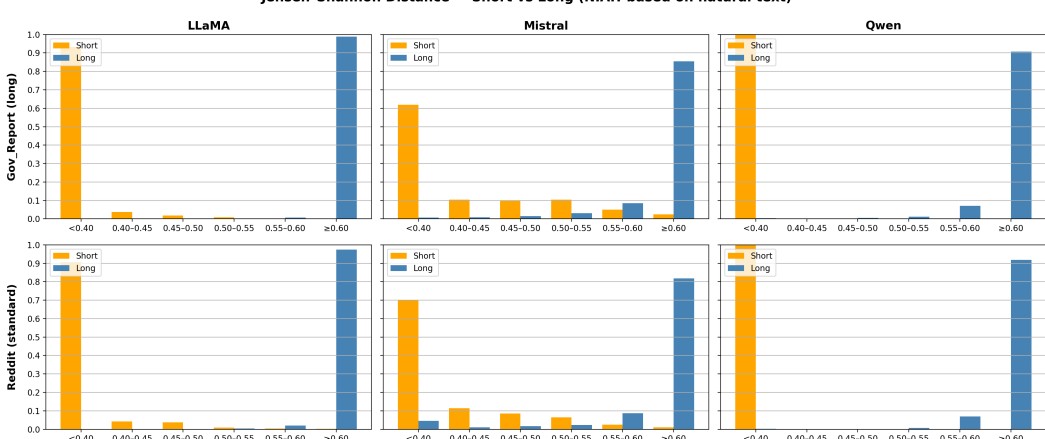

Figure 22: **NIAH-based experiment on General text result:** LSDS distribution across three models (LLaMA, Mistral, Qwen) on long-context dataset (GovReport) and standard-length context (Reddit), each with 10000 samples. Each subplot shows the proportion of tokens falling into fixed JSD bins.

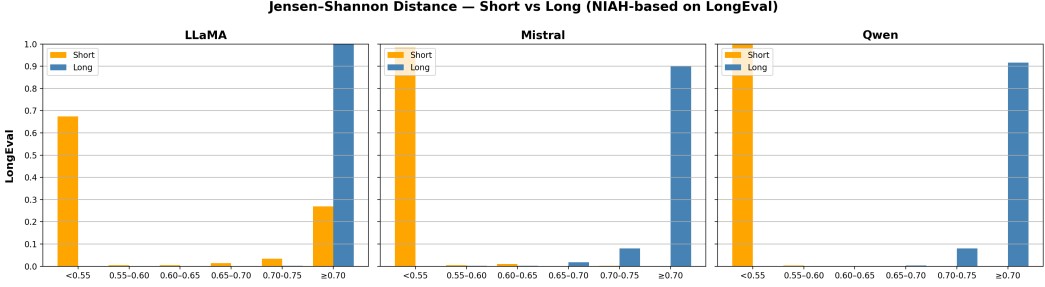

Figure 23: **Controlled validation experiment on LongEval task result:** LSDS distribution across three models (LLaMA, Mistral, Qwen) on the LongEval synthetic benchmark. Each subplot shows the proportion of tokens falling into fixed JSD bins.

queries we use a *<REGISTER_CONTENT>* from a recent line. This allows us to have a more direct comparison between long and short context as the nature of both tokens are the same (same style of registered content as opposed to comparing to line-id) while the positions of the reference line in the local or long context position changes their LSDS behavior.

the only other difference compared to our previous setup is how here, we consider the last 64 tokens as local context, only to ensure that a full line from a local context will always fit inside the short context window. Results are provided in Fig. 24 and illustrate how this setup complies with our previous observations.

### E.1.4 PRIOR-WORK ORACLE (LSD/LCL) ON GENERAL TEXT

As described in Section 5, we evaluate a *prior-work oracle* detector that follows the log-probability intervention from Fang et al. (2025). The oracle Long-Short Difference (LSD) and Long-Context Likelihood (LCL) are defined as follows:
For sequence $\mathbf{s}$ with corpus next-token $t$, and a language model $P_\theta$,

$$\text{LSD}_\theta(\mathbf{s}|t) = \log\left[\mathbf{p}(\mathbf{s})\right]_t \ - \ \log\left[\mathbf{p}(\mathbf{s}_{[-32:]})\right]_t \tag{2}$$

and

$$\text{LCL}_\theta(\mathbf{s}|t) = \log\left[\mathbf{p}(\mathbf{s})\right]_t \tag{3}$$

Figure 25 shows the per-model distributions of LSDS for LLaMA-3-8B, Mistral-7B-Instruct, and Qwen2-7B on the same data used in the main body. The per-model trends mirror the pooled result:

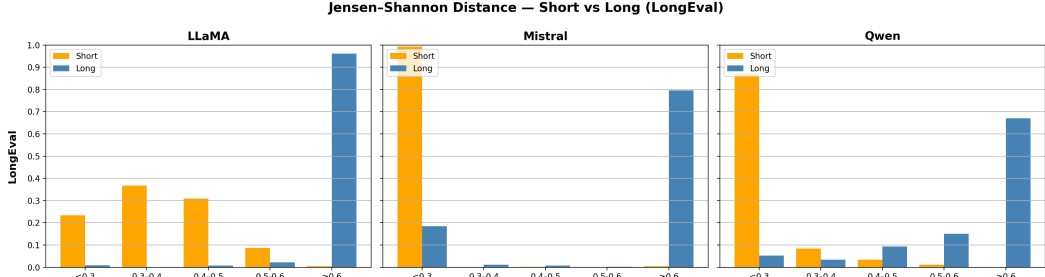

Figure 24: **Controlled validation experiment on LongEval task result with local lines as short-context**: similar results to that presented in 23 except here we use the *<REGISTER_CONTENT>* from local lines as the local context token instead of *line_id*.

using a threshold of $\tau = 0.6$, we find consistently that $80 - 90\%$ of oracle-labeled long-context sequences achieve $\text{LSDS}(\mathbf{s}) \geq 0.6$, while fewer than $5 - 10\%$ of short-context sequences exceed this threshold.

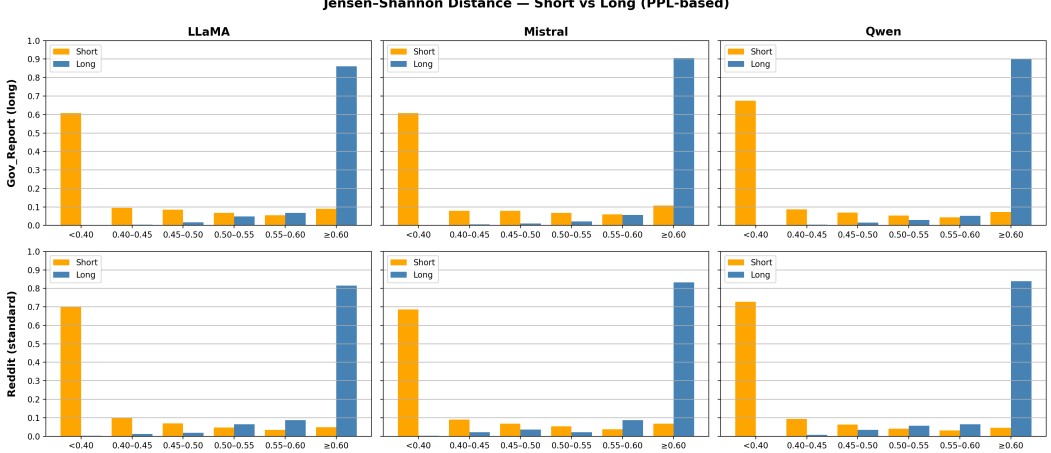

Figure 25: **Prior-work oracle (LSD/LCL) on general text result:** LSDS distribution across three models (LLaMA, Mistral, Qwen) on long-context dataset (GovReport) and standard-length context (Reddit), each with 10000 samples. Each subplot shows the proportion of tokens falling into fixed JSD bins.

### E.1.5 MCL ORACLE ON GENERAL TEXT

Additionally, we evaluate our *MCL Oracle* on general text. Following the definition and computation procedure in Section 3, we compute the MCL of each given context. Tokens with small MCL values ($\leq 32$) are classified as short-context, while those requiring larger suffixes ($> 32$) are classified as long-context. A limitation of this approach is that the MCL is not valid for every token, as MCL only exists for tokens which the model predicts correctly and confidently. Thus we can only perform our analysis on context/token pairs which the model performs well on.

We then compare the JSD distributions of tokens across the two categories. Figure 27 shows that the detected trend is still preserved.

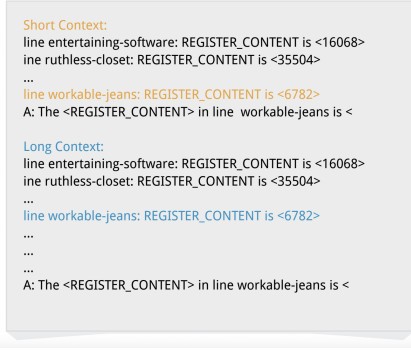

Figure 26: An example of another synthetic setup on the LongEval benchmark.

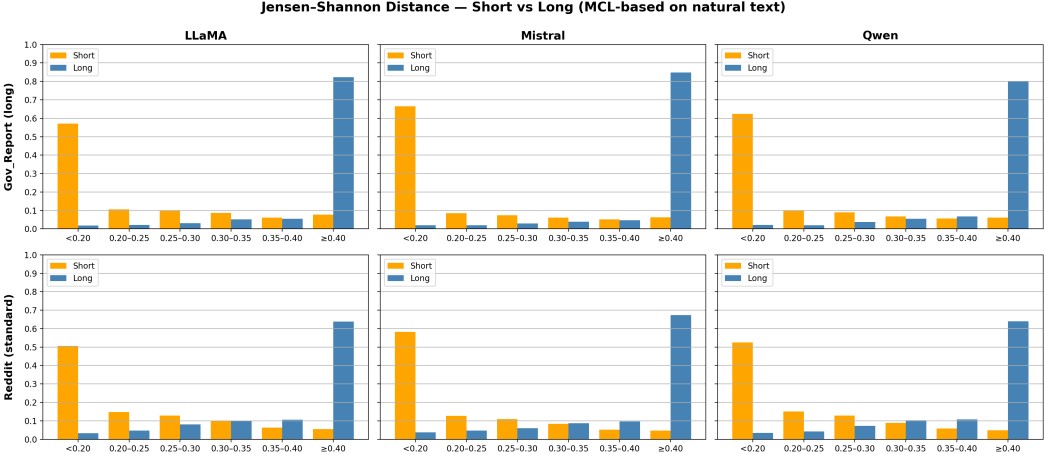

Figure 27: **MCL oracle on general text result:** LSDS distribution across three models (LLaMA, Mistral, Qwen) on long-context dataset (GovReport) and standard-length context (Reddit), each with 10000 samples. Each subplot shows the proportion of tokens falling into fixed JSD bins.

### E.1.6 QUANTIFYING LSDS SEPARATION

In addition to the clear visual separation observed in the LSDS distributions for short vs. long contexts, we also report the ROC area under the curve (AUC), which provides a single quantitative measure of binary separability. The ROC curve captures the trade-off between the true-positive and false-positive rates when detecting long-context usage across a range of LSDS thresholds $\epsilon$. A higher AUC corresponds to stronger separability between the LSDS values under long vs. short contexts, thereby validating LSDS as a reliable signal for distinguishing context usage.

Figure 28 (a) shows the resulting AUC values on the GovReport dataset using LLaMA with LSD+LCL as the oracle. We additionally report results on the LongEval task (b,c), where short/long-contexts are formed as examples in Figure 26. Similar trends were observed across other datasets and models but are omitted here for brevity.

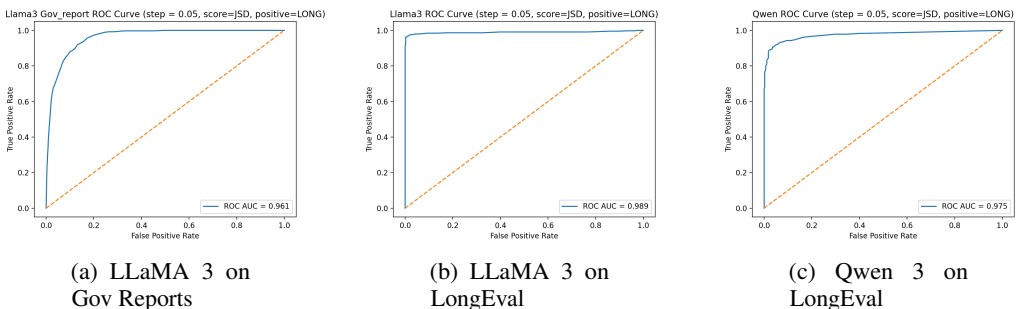

(a) LLaMA 3 on Gov Reports

(b) LLaMA 3 on LongEval

(c) Qwen 3 on LongEval

Figure 28: **AUC for LSDS:** ROC area under curve for LSDS accuracy for capturing the long contexts. For **(a)** we use the real data with LCL+LSD as oracle and for **(b,c)** we use the LongEval dataset.

### E.2 ABLATION ON NUCLEUS SAMPLING PARAMETER ($p$)

In this section, we examine how varying the nucleus sampling parameter $p$ influence our LSDS and the resulting JSD threshold. We report results for 4 settings: no nucleus sampling, $p = 0.95$, $p = 0.90$, and $p = 0.80$. We conduct the experiment with Mistral-7B-Instruct (v0.2) (Jiang et al., 2023) on 5000 contexts randomly selected from Government Report (Huang et al., 2021), with full context length ranging from 100–1000 tokens. The ground truth oracle we use here is *Prior Work Oracle* as introduced in Sec. 5.

Across all settings, we observe that the separation between short- and long-context distribution is preserved (see Figure 29). Regardless of $p$, short-context examples consistently concentrate in low JSD bins ($< 0.40$), while long-context examples dominate the high JSD bins ($\geq 0.60$). However, the optimal detection thresholds vary slightly depending on $p$ value.

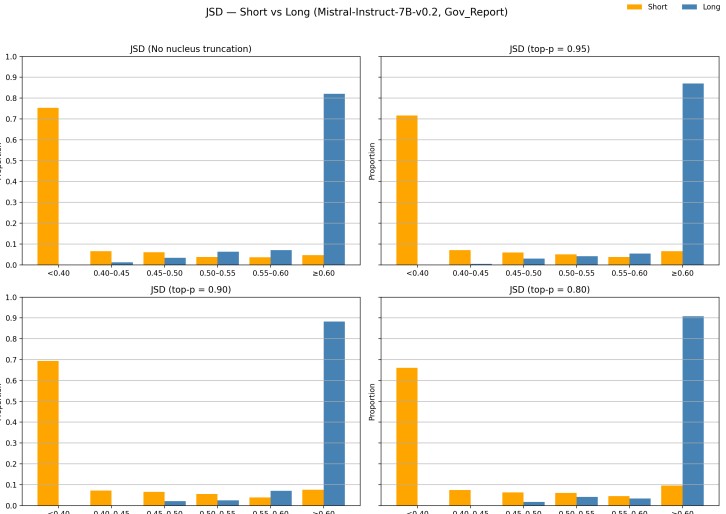

Figure 29: LSDS distribution with Mistral-7B-Instruct-v0.2 on GovReport(5000 samples) with no nucleus sampling, $p = 0.95$, $p = 0.90$, and $p = 0.80$.

To quantitatively assess performance, we compute the maximized Youden's $J$ statistic (Youden, 1950):

$$J = \max_\theta \left\{ \text{TPR}(\theta) - \text{FPR}(\theta) \right\}$$

where TPR is the true positive rate and FPR is the false positive rate, and $\theta$ is the decision threshold. Unlike accuracy, which can be biased by imbalanced class distributions, $J$ simultaneously accounts for both sensitivity (true positive rate) and specificity (true negative rate). In our case, detecting long- versus short-context tokens requires balancing correct identification of long-context samples with avoidance of false detections. Thus Youden's $J$ provides an appropriate criterion for threshold selection. Table 5 summarizes the best thresholds and corresponding performance metrics.

Table 5: Ablation study on different nucleus sampling parameters ($p$). Reported are the optimal threshold, maximized Youden's $J$, true positive rate (TPR), false positive rate (FPR), precision, recall, and accuracy.

| Case | Threshold | $J$ | TPR | FPR | Precision | Recall | Accuracy |
|------|-----------|-----|-----|-----|-----------|--------|----------|
| No nucleus | 0.49 | 0.834 | 0.967 | 0.133 | 0.268 | 0.967 | 87.2% |
| top-$p$=0.95 | 0.53 | 0.831 | 0.950 | 0.119 | 0.286 | 0.950 | 88.4% |
| top-$p$=0.90 | 0.55 | **0.840** | 0.954 | 0.114 | 0.296 | 0.954 | 88.9% |
| top-$p$=0.80 | 0.62 | 0.816 | 0.895 | 0.079 | 0.361 | 0.895 | **91.9%** |

The setting $p = 0.90$ achieves the highest Youden's $J$ (0.840), indicating the most balanced trade-off between sensitivity (TPR) and specificity ($1 - \text{FPR}$). While $p = 0.80$ achieves the highest accuracy (91.9%), its $J$ value is lower, reflecting weaker overall discriminative balance. Therefore, we adopt $p = 0.90$ as our standard nucleus sampling configuration in the main experiments. This choice is consistent with prior work recommending $p$ in the 0.9–0.95 range for stable yet diverse generation quality (Holtzman et al., 2020).

### E.3    ABLATION ON SHORT-CONTEXT LENGTH

We investigate the effect of varying the short-context prefix length $\ell$ on LSDS and JSD threshold. Experiments are run with Mistral-7B-Instruct-v0.2 on Government Report, using 5000 randomly selected samples, with full context length ranging from 100–1000 tokens. The ground truth oracle we use here is *Prior Work Oracle* as introduced in Sec. 5, and the short prefix length in Equation 2 is adjusted accordingly when computing LSD.

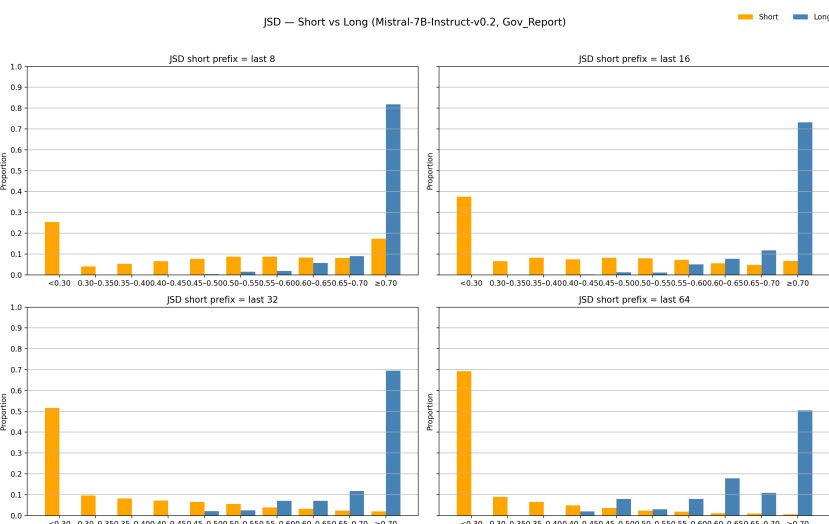

Figure 30: LSDS distribution with Mistral-7B-Instruct-v0.2 on GovReport(5000 samples) with short prefix length $\ell = 8, 16, 32, 64$.

**Different fixed short-prefix length.**    Figure 30 shows results for $\ell = 8, 16, 32, 64$. We observe that the overall distributional separation between short and long contexts is preserved across all choices. However, very small short-context prefixes (e.g., $\ell = 8$) produce more imbalanced distributions: nearly 20% of short-context tokens fall above the detection threshold. This indicates that while separation is robust, extremely small short prefixes may introduce higher false-positive rates. In contrast, $\ell = 32, 64$ yield clearer separation, with false negative rates less than 5%, suggesting they are preferable for stable detection. Although using longer prefixes incurs slightly higher computational cost, the improved robustness makes the tradeoff worthwhile—especially in applications where false positives are costly.

**Adaptive Short-Context Length.**    We also eval­uate an adaptive strategy where the short-context length is chosen as a proportion of the full se­quence length, $\ell = 0.1|s|$. Figure 31 demon­strates consistent separation trends under this setup, while adapting naturally to dataset-specific context lengths. Such adaptive schemes may pro­vide better cross-dataset generalization, although their computational implications require further study. We leave exploration of such direction for future work.

Figure 31: LSDS distribution with Mistral-7B-Instruct-v0.2 on GovReport(5000 samples) with short prefix length $\ell = 0.1|s|$, consistent separa­tion of the distributions is preserved.

### E.4    COMPUTATIONAL OVERHEAD OF LSDS DETECTION AND TABOO

In this section, we quantify the computational cost introduced by LSDS. Recall that LSDS requires evaluating token distributions under both the full context and a short suffix (e.g., 32 tokens). For a

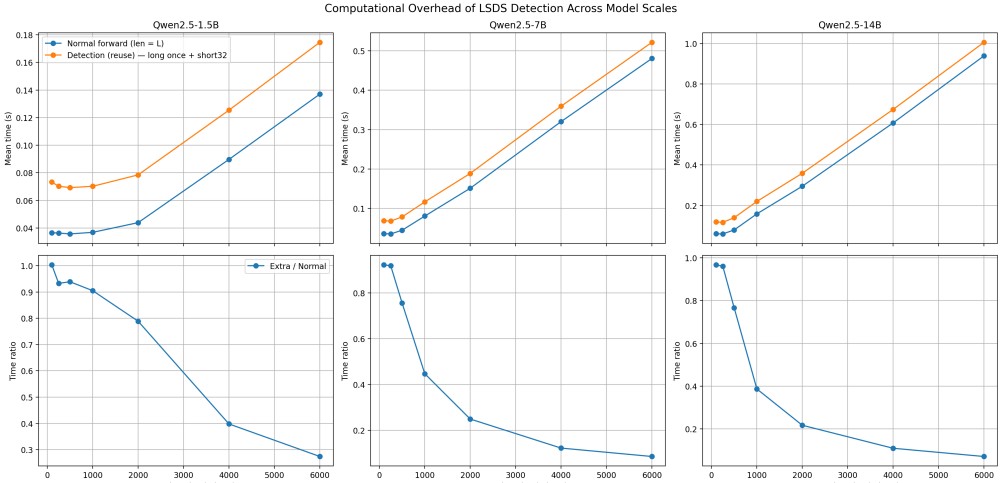

Figure 32: Computational overhead of LSDS detection across Qwen2.5 models (1.5B, 7B, 14B). Top row: mean forward time for normal inference versus detection with reuse (long context + 32-token suffix). Bottom row: relative overhead, measured as the ratio of additional cost to normal forward time. The added cost is nearly constant, demonstrating that LSDS overhead is limited, predictable, and diminishes in importance as sequence length grows.

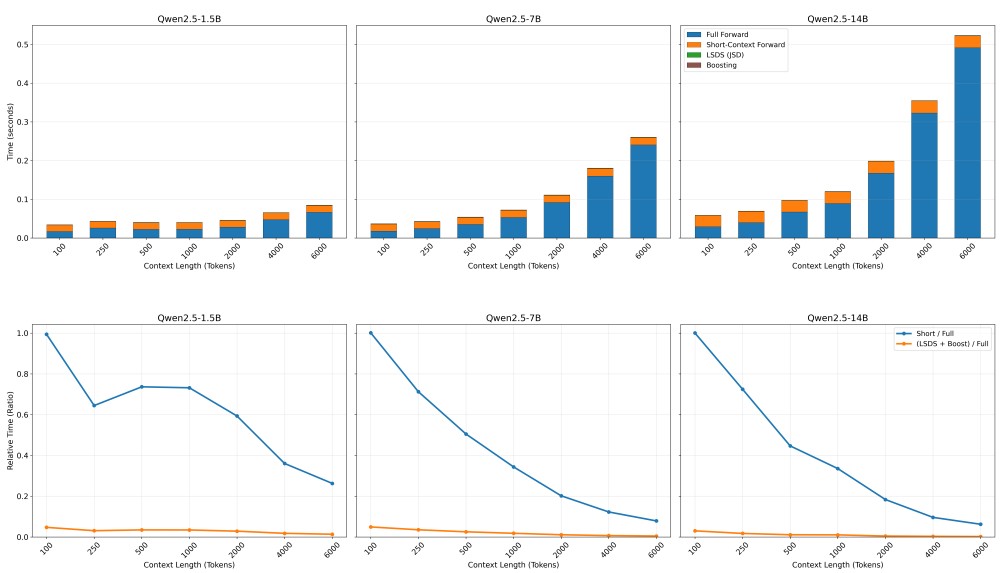

Figure 33: Comprehensive TaBoo runtime analysis. **Top:** absolute runtime of each component (full forward, short-context forward, LSDS, and boosting). **Bottom:** relative cost ratios showing that the LSDS and boosting computations are negligible compared to the full forward pass. Additionally the additionally short-context pass becomes relatively cheaper with larger models and longer context.

generation task, the full forward is already computed, so the incremental cost reduces to a single short forward (32 tokens) plus lightweight top-p filtering and JSD computation.

We benchmark this overhead on Qwen2.5 models of different scales (1.5B, 7B, 14B) using contexts $|\mathbf{s}| = 100, 250, 500, 1000, 2000, 4000, 6000$. The absolute overhead remained nearly constant: about 35–40 ms for 1.5B, 37–41 ms for 7B, and 58–67 ms for 14B. Relative cost is high at short contexts ($> 90\%$ at $|\mathbf{s}| = 100$) but drops to $\approx 6 - 8\%$ at $|\mathbf{s}| = 6000$. These results confirm that LSDS adds only minor and predictable overhead with cache reuse.

For other purposes or as a conservative upper bound, one can rerun the full forward pass, which would roughly double the cost. In addition, using adaptive short-prefix length as described above introduces overhead that scales linearly with context length.

Figure 32 illustrates the relative computational cost of a full forward pass compared with the extra short-context forward required for detection across all three models. Regarding the calculations for LSDS and selecting target tokens to boost, the process selection takes on average less than $\leq 1\%$ of the computation time in comparison to the forward pass (see Fig . 33), thus we consider it as negligible.

# F  LONG-CONTEXT TOKEN DETECTION AND BOOSTING

## F.1  PROBABILITY SHIFT AS SELECTION METRIC

For our selection of long context relevant tokens, we decide to to use the change in the probability values as our determining metric. While we decide on using the probability difference LSPS $(t|\mathbf{s}) = \mathbf{p}_{p=0.9}(\mathbf{s})_t - \mathbf{p}_{p=0.9}(\mathbf{s}_{[-32:]})_t$ as the determining factor, alternative choice could be the use of probability ratios rather than the difference. In other words we can define **Long-Short Probability Ratio**:

$$\mathsf{LSPR}\,(t|\mathbf{s}) = \log(\frac{\mathbf{p}_{p=0.9}(\mathbf{s})_t}{\mathbf{p}_{p=0.9}(\mathbf{s}_{[-32:]})_t})$$

A comparison between this form of metric and our probability difference **LSPS** becomes important when you consider works such as Fang et al. (2025) use the log probability ratio to detect long-short context/next-tokens. Additionally, in Malkin et al. (2022) and Duh et al. (2024) the authors use the log ratio for probability and logits with in the form of $\mathbf{p}(\mathbf{s})_t^{1.5}\mathbf{p}_{[-32:]}(\mathbf{s})_t^{0.5}(t)$ to improve the final probability of long-context tokens. Implicitly, such methods select and increase the probability of tokens who's probability ratio increases rather than the difference itself.

In order to compare the choice of metric, we compare the use of **LSPS** and **LSPR** when boosting the final word/token of on the validation set of LAMBADA (Paperno et al., 2016) dataset. For each context, we have a ground truth next token, which we call the target token $\hat{t}$. All other tokens we call incorrect tokens. We analyze the choice of which token's are selected for probability increase rather than the improvement itself, therefore we don't include multiplicative factors like $\lambda$. For each case of boosting, we consider the following scenarios:

- Scenario 1 (Best): $\hat{t} \in \mathcal{B}$ and $\mathbf{p}_{p=0.9}(\mathbf{s})_{\hat{t}} \geq \mathbf{p}_{p=0.9}(\mathbf{s})_t$ for any other $t \in \mathcal{B}$. Under such a scenario, even if some non answer tokens are selected for probability increase, if we grow $\lambda$ eventually greedy sampling would select select the answer token.

- Scenario 2 (Bad): $\hat{t} \in \mathcal{B}$ however, there exists at least one token $t \in \mathcal{B}$ with a higher probability than the answer token $\mathbf{p}_{p=0.9}(\mathbf{s})_{\hat{t}} \geq \mathbf{p}_{p=0.9}(\mathbf{s})_t \ \forall \ t \in \mathcal{B}$. Under such a scenario, while a higher $\lambda$ value increases the probability of $\hat{t}$ and thus improving perplexity, but there there isn't a possibility of the ground token to be selected by a greedy sampler.

- Scenario 3 (Worst): $\hat{t} \notin \mathcal{B} \neq \varnothing$. In this case we didn't detect the ground truth token but rather chose a set of incorrect tokens. Under such circumstances, the model next token perplexity would suffer.

- Scenario 4 (Neutral): $\mathcal{B} = \varnothing$ and so no change is made to the probability distribution.

We use nucleus sampling with $p = 0.9$ and set a minimum probability of $10^{-6}$ as the minimum, in order to prevent numerical instability for the log probability calculations. We will select a range of $\epsilon$ values as the selection threshold with respect to the metrics themselves. We report the ratio of each of above's scenarios from the 5153 examples in the LAMBADA validation dataset.

We observe that using log probability ratio is more likely to lead to a wrong token selection for the boosted set without including the target token $\hat{t}$. This case is particularly dangerous as it negatively impacts a metric such as perplexity. Additionally, results suggest that the selection performance highly sensitive to the choice of $\epsilon$.

Table 6: Analysis of ground truth $\hat{t}$ token being selected for probability improvement. A comparisong between LSPD (diff) and LSPR (log ratio).

(a) LSPD (diff case)

| $\epsilon$ | Best | Bad | Worst | Neutral |
|---|---|---|---|---|
| 0.04 | 66.8% | 10.0% | 14.2% | 9.1% |
| 0.06 | 65.5% | 7.8% | 12.8% | 13.9% |
| 0.08 | 64.3% | 6.4% | 11.1% | 18.2% |
| 0.10 | 62.9% | 5.4% | 9.5% | 22.1% |
| 0.12 | 61.4% | 4.4% | 8.6% | 25.7% |
| 0.14 | 59.8% | 3.6% | 7.7% | 28.9% |
| 0.16 | 58.1% | 3.0% | 6.8% | 32.1% |
| 0.18 | 56.5% | 2.4% | 6.1% | 35.0% |
| 0.20 | 55.0% | 2.0% | 5.6% | 37.5% |
| 0.22 | 53.3% | 1.5% | 5.1% | 40.1% |

(b) LSPR (log ratio case)

| $\epsilon$ | Best | Bad | Worst | Neutral |
|---|---|---|---|---|
| 0.5 | 60.3% | 10.7% | 20.2% | 8.8% |
| 1.0 | 57.7% | 7.2% | 22.1% | 13.0% |
| 1.5 | 56.2% | 5.2% | 23.1% | 15.4% |
| 2.0 | 54.8% | 4.3% | 23.9% | 17.0% |
| 2.5 | 53.2% | 3.7% | 24.8% | 18.3% |
| 3.0 | 51.7% | 3.1% | 25.8% | 19.5% |
| 3.5 | 49.7% | 2.6% | 27.5% | 20.3% |
| 4.0 | 46.8% | 2.0% | 29.4% | 21.8% |
| 4.5 | 43.2% | 1.6% | 31.5% | 23.7% |
| 5.0 | 38.5% | 1.4% | 34.2% | 26.0% |

From our analysis such observations can be explained by the sensitivity of log ratio. Particularly for tokens with near zero probability, small perturbation when moving from short to long context probability can be registered under large change in probability ratio changes. Consider using $\log_{10}$ for this explanation only (experiments are done with natural log). If a token's probability goes from $\mathbf{p}_{p=0.9}(\mathbf{s}_{[-32:]})_t = 10^{-6} \rightarrow \mathbf{p}_{p=0.9}(\mathbf{s})_t = 10^{-2}$, the log probability ratio is set to $4$. Similarly if the token's probability goes from $10^{-3} \rightarrow 10$ we have a similar probability ratio change. However, for the first scenario the full context token probability is near zero, while for the second case the token has a 10% probability. If we go with a the log ratio, both these tokens are treated equally.

On the other hand, for the probability difference case, we have implicitly set a minimal lower bound on the $\mathbf{p}_{p=0.9}(\mathbf{s})_t \geq \epsilon$ by only selecting tokens where $|\mathbf{p}_{p=0.9}(\mathbf{s})_t - \mathbf{p}_{p=0.9}(\mathbf{s}_{[-32:]})_t| \geq \epsilon$. This allows us to circumvent tokens who have very low probability where the log ratio behavior could be noisy and misleading. We believe this is a potential reason for why the long context token selection algorithm in Fang et al. (2025) uses $\log [\mathbf{p}(\mathbf{s})]_t \geq 10^{-2}$ ( **Long-Context Likelihood (LCL)** ) in addition to their log probability ratio to classify tokens.

While we find probability difference to be a safer and more effective option for our studies, further analysis could help discover new methods for ad-hoc detection of long-context relevant token by comparing short-long context distributions. We believe this can serve a direction for future research.

Table 7: XSum summarization results (Average over all generations; Best-per-example in parentheses). Bold = best Average, Underline = best Best-per-example. Regarding the low performance of CAD, we employ the same methods used for QA with a $\alpha = 0.5$ hyperparameter selection similar to their setup. Interestingly, our average Vanilla score is similar to theirs. Nevertheless, this table is provided to show that our TaBoo method effectively removes short-context bias in summarization, not to compare our methods directly with CAD.

| Model | Method | F1 | BLEU | ROUGE-L |
|---|---|---|---|---|
| | Vanilla | 18.1 (28.6) | 2.3 (5.2) | 17.4 (27.3) |
| LLaMA-2-7B | CAD | 10.4 (16.4) | 1.0 (1.6) | 10.0 (15.5) |
| | TaBoo | **21.0** (31.5) | **3.0** (6.9) | **19.9** (30.0) |
| | Vanilla | 18.4 (29.6) | 2.4 (5.9) | 17.8 (28.7) |
| LLaMA-3-8B | CAD | 12.0 (18.6) | 1.1 (2.0) | 11.5 (17.5) |
| | TaBoo | **20.7** (31.8) | **3.0** (6.9) | **19.7** (30.5) |
| | Vanilla | 18.9 (26.5) | 2.4 (4.5) | 17.9 (25.1) |
| Mistral-8B | CAD | 9.4 (12.8) | 0.8 (1.1) | 9.2 (12.3) |
| | TaBoo | **21.3** (28.0) | **3.1** (5.5) | **20.4** (26.7) |
| | Vanilla | 20.8 (28.3) | 2.7 (5.0) | 19.5 (26.7) |
| Qwen2-7B | CAD | 12.0 (17.3) | 1.1 (1.7) | 11.5 (16.2) |
| | TaBoo | **21.1** (27.9) | **2.7** (4.8) | **19.9** (26.2) |
| | Vanilla | 24.4 (33.5) | 3.9 (8.0) | 22.4 (31.4) |
| Mistral-Inst-8B | CAD | 12.2 (18.4) | 1.2 (2.1) | 11.3 (17.0) |
| | TaBoo | **25.2** (33.1) | **4.4** (7.9) | **23.3** (31.0) |

F.2    TABOO ALGORITHM

Algorithm 1 presents our TaBoo method.

---
**Algorithm 1:** Long Context Token Boosting with JSD and Nucleus Constraint

---
**Input:** Short- and full-context distributions $\mathbf{p}_\phi(\mathbf{s}_{[-32:]})$, $\mathbf{p}_\phi(\mathbf{s})$; thresholds $\epsilon$, $\gamma$; boost factor $\lambda$;
  decoding $\phi$: nucleus sampling with parameter $p$; support set $\mathcal{V}^{\text{nuc}}$ of $\mathbf{p}_\phi(\mathbf{s})$
**Output:** TaBoo distribution $\widetilde{\mathbf{p}}_\phi(\mathbf{s})$

1   Initialize (to the raw probability distribution): $\widetilde{\mathbf{p}}(\mathbf{s}) \leftarrow \mathbf{p}_\phi(\mathbf{s})$ ;
2   **Step 1:** Identify whether sequece is long-context:
3   **if** $\mathsf{LSDS}(\mathbf{s}) \leq \gamma$ **then**
4    **return** $\mathbf{p}_\phi(\mathbf{s})$ ;      // Return unmodified nucleus distribution
5   **Step 2:** Identify set of long-context-relevant tokens: $\mathcal{B} \leftarrow \{t \in \mathcal{V}^{\text{nuc}} \mid \mathsf{LSPS}(t|\mathbf{s}) > \epsilon\}$ ;
6   **Step 3:** Targeted Boosting: **foreach** *token* $t \in \mathcal{B}$ **do**
7    $[\widetilde{\mathbf{p}}(\mathbf{s})]_t \leftarrow \lambda \cdot [\mathbf{p}_\phi(\mathbf{s})]_t$ ;
8   Re-normalize and apply nucleus decoding: $\widetilde{\mathbf{p}}(\mathbf{s}) \overset{\text{nuc}}{\leftarrow} \widetilde{\mathbf{p}}(\mathbf{s})/\operatorname{sum}(\widetilde{\mathbf{p}}(\mathbf{s}))$ for all $t \in \mathcal{V}$ ;
9   **return** $\widetilde{\mathbf{p}}(\mathbf{s})$

---

### F.3   AVERAGE PERFORMANCE WITH STANDARD ERRORS

We report the Average scores for F1, BLEU, and ROUGE-L across all generations; values in parentheses are the corresponding standard errors (SE). These SEs capture variability across evaluation examples on the same split. Results for LLaMA-2-7B on MultifieldQA-en are omitted due to the 4096-token context limit.

Table 8: F1, BLEU, and ROUGE-L (Average; standard error in parentheses). Bold = best Average within each dataset block.

| Model | Method | NarrativeQA | | | HotpotQA | | | MultifieldQA-en | | |
|---|---|---|---|---|---|---|---|---|---|---|
| | | F1(↑) | BLEU(↑) | ROUGE-L(↑) | F1(↑) | BLEU(↑) | ROUGE-L(↑) | F1(↑) | BLEU(↑) | ROUGE-L(↑) |
| LLaMA-2-7B | Vanilla | 16.1 (±0.29) | 2.9 (±0.09) | 22.4 (±0.36) | 25.2 (±0.48) | 7.7 (±0.21) | 32.5 (±0.53) | NA | NA | NA |
| | CAD | 22.5 (±0.40) | 4.3 (±0.12) | 30.7 (±0.48) | 28.3 (±0.53) | 8.5 (±0.24) | 34.3 (±0.56) | NA | NA | NA |
| | TaBoo | **24.1** (±0.41) | **4.9** (±0.14) | **31.7** (±0.47) | **32.8** (±0.58) | **10.3** (±0.27) | **39.6** (±0.61) | NA | NA | NA |
| LLaMA-3-8B | Vanilla | 24.0 (±0.37) | 4.9 (±0.13) | 32.7 (±0.46) | 29.2 (±0.49) | 9.0 (±0.23) | 41.6 (±0.58) | 9.9 (±1.62) | 7.7 (±1.17) | 24.0 (±1.73) |
| | CAD | **35.4** (±0.52) | **7.3** (±0.18) | **49.4** (±0.62) | 27.7 (±0.53) | 8.5 (±0.23) | 46.9 (±0.68) | 18.8 (±1.72) | 6.4 (±1.03) | 26.2 (±2.02) |
| | TaBoo | 32.0 (±0.47) | 7.2 (±0.18) | 42.3 (±0.53) | **33.1** (±0.55) | **10.6** (±0.26) | **48.1** (±0.64) | **21.9** (±1.74) | **9.1** (±1.29) | **28.2** (±1.87) |
| Mistral-7B-v0.1 | Vanilla | 25.7 (±0.41) | 5.1 (±0.13) | 33.7 (±0.48) | 33.0 (±0.54) | 10.1 (±0.24) | 43.1 (±0.59) | 20.6 (±1.54) | 6.9 (±1.07) | 26.0 (±1.72) |
| | CAD | 34.3 (±0.55) | 7.1 (±0.17) | 43.1 (±0.57) | 35.9 (±0.62) | 11.0 (±0.28) | 41.6 (±0.63) | 18.8 (±1.44) | 5.9 (±0.99) | 24.9 (±1.65) |
| | TaBoo | **35.3** (±0.52) | **7.7** (±0.15) | **44.4** (±0.46) | **37.1** (±0.61) | **11.7** (±0.29) | **46.3** (±0.64) | **23.0** (±1.65) | **8.6** (±1.27) | **29.5** (±1.83) |
| Qwen2-7B | Vanilla | 33.6 (±0.49) | 8.1 (±0.19) | 42.4 (±0.53) | 59.4 (±0.64) | 20.1 (±0.36) | 62.9 (±0.63) | 31.1 (±1.86) | 15.1 (±1.63) | 41.3 (±2.01) |
| | CAD | 36.6 (±0.58) | 8.8 (±0.22) | 45.3 (±0.60) | 59.3 (±0.68) | 20.5 (±0.39) | 62.2 (±0.67) | 30.6 (±1.80) | 14.0 (±1.49) | 40.0 (±2.10) |
| | TaBoo | **38.5** (±0.56) | **9.7** (±0.15) | **48.2** (±0.64) | **63.2** (±0.67) | **21.7** (±0.39) | **66.6** (±0.65) | **32.3** (±1.85) | **15.3** (±1.64) | **42.3** (±2.07) |

### F.4   IMPACT OF DECODING

Another question we ask is the impact of using decoding methods when implementing the token selection during Taboo. In order to test this out, we without any nucleus sampling and using the raw pr

Results are provided in Tab . 9. Comparing results when using LSPS, we don't observe a major change, one case being better than the other depending on the value of $\epsilon$. For LSPR on the other hand, we can see that not doing any nucleus sampling leads to much more **Worst** case scenarios where we don't boost the answer token but rather a irrelevant one. This observation both points towards the robustness of LSPS w.r.t the next token probability distribution (with or without decoding). Additionally it points our the potential problem with using the log ratio of probabilities.

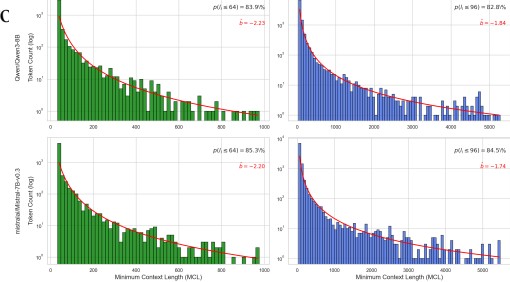

Figure 34: MCL Results on more recent models.

Table 9: Similar setup to Tab. 6 but this time without any nucleus sampling.

<table>
<tr><td colspan="5" align="center">(a) LSPD (diff case)</td><td colspan="5" align="center">(b) LSPR (log ratio case)</td></tr>
<tr><td>$\epsilon$</td><td>Best</td><td>Bad</td><td>Worst</td><td>Neutral</td><td>$\epsilon$</td><td>Best</td><td>Bad</td><td>Worst</td><td>Neutral</td></tr>
<tr><td>0.04</td><td>66.7%</td><td>9.3%</td><td>13.8%</td><td>9.1%</td><td>0.5</td><td>60.6%</td><td>11.2%</td><td>28.1%</td><td>0.1%</td></tr>
<tr><td>0.06</td><td>65.4%</td><td>7.4%</td><td>12.0%</td><td>15.2%</td><td>1.0</td><td>58.5%</td><td>7.5%</td><td>34.6%</td><td>0.3%</td></tr>
<tr><td>0.08</td><td>63.9%</td><td>5.9%</td><td>10.2%</td><td>20.0%</td><td>1.5</td><td>57.0%</td><td>5.4%</td><td>38.0%</td><td>0.4%</td></tr>
<tr><td>0.10</td><td>62.0%</td><td>4.9%</td><td>8.7%</td><td>24.0%</td><td>2.0</td><td>55.6%</td><td>4.3%</td><td>40.4%</td><td>0.5%</td></tr>
<tr><td>0.12</td><td>60.3%</td><td>3.9%</td><td>7.7%</td><td>28.1%</td><td>2.5</td><td>54.0%</td><td>3.6%</td><td>42.7%</td><td>0.6%</td></tr>
<tr><td>0.14</td><td>58.6%</td><td>3.2%</td><td>6.6%</td><td>31.7%</td><td>3.0</td><td>52.4%</td><td>3.0%</td><td>45.0%</td><td>0.7%</td></tr>
<tr><td>0.16</td><td>56.9%</td><td>2.6%</td><td>5.9%</td><td>34.7%</td><td>3.5</td><td>50.3%</td><td>2.4%</td><td>47.9%</td><td>0.7%</td></tr>
<tr><td>0.18</td><td>55.2%</td><td>2.1%</td><td>5.2%</td><td>37.6%</td><td>4.0</td><td>47.3%</td><td>1.7%</td><td>51.6%</td><td>0.8%</td></tr>
<tr><td>0.20</td><td>53.6%</td><td>1.6%</td><td>4.8%</td><td>40.9%</td><td>4.5</td><td>43.8%</td><td>1.3%</td><td>56.4%</td><td>0.9%</td></tr>
<tr><td>0.22</td><td>51.9%</td><td>1.2%</td><td>4.3%</td><td>42.6%</td><td>5.0</td><td>39.0%</td><td>1.0%</td><td>61.7%</td><td>0.9%</td></tr>
</table>

### F.5 ANALYSIS ON RECENT DATASETS/MODELS

We provide additional MCL results on more recent models, including Qwen-3-8B Base (Yang et al., 2025) and Mistral-v0.3-7B (Mistral AI, 2024), evaluated on both short- and long-context datasets. These results, shown in Fig. 34, confirm that short-context dominance persists even in these updated model families. We further evaluate TaBoo on two additional QA datasets from L-Eval (An et al., 2023), namely *NaturalQuestions*, *MultiDocQA* and *ScientificQA*, using the same models. To ensure that the QA tasks genuinely require contextual information (i.e., cannot be answered in a closed-book setting), we first run each model on the question alone and remove any example for which the model achieves an **F1** score of $\geq 0.4$. As shown in Tab. 10, both models find these QA tasks challenging due to their longer contexts and higher difficulty. Nonetheless, TaBoo consistently improves performance over vanilla inference and CAD, demonstrating that our boosting approach remains effective even on these stronger, more context-heavy benchmarks.

Table 10: F1, BLEU, and ROUGE-L (Average; standard error in parentheses) on more recent models. Bold = best Average within each dataset block.

| Model | Method | NaturalQuestions | | | MultiDocQA | | | ScientificQA | | |
|---|---|---|---|---|---|---|---|---|---|---|
| | | F1(↑) | BLEU(↑) | ROUGE-L(↑) | F1(↑) | BLEU(↑) | ROUGE-L(↑) | F1(↑) | BLEU(↑) | ROUGE-L(↑) |
| Qwen3-8B | Vanilla | 27.10 (±1.99) | 7.69 (±0.80) | 33.99 (±2.16) | 15.35 (±0.60) | 2.85 (±0.34) | 18.25 (±0.68) | 16.15 (±0.82) | 4.09 (±0.40) | 24.71 (±0.94) |
| | CAD | 13.84 (±1.17) | 3.21 (±0.48) | 19.93 (±1.44) | 10.36 (±0.57) | 1.61 (±0.26) | 13.39 (±0.70) | 13.93 (±0.84) | 4.40 (±0.46) | 19.38 (±0.96) |
| | TaBoo | **32.89** (±2.18) | **11.30** (±1.18) | **38.56** (±2.29) | **16.87** (±0.65) | **3.32** (±0.36) | **20.40** (±0.73) | **21.17** (±0.91) | **6.51** (±0.54) | **30.68** (±1.01) |
| Mistral-0.3 | Vanilla | 15.78 (±1.17) | 3.27 (±0.44) | 17.90 (±1.23) | 9.97 (±0.51) | 1.98 (±0.33) | 10.49 (±0.52) | 12.48 (±0.61) | 2.33 (±0.23) | 16.73 (±0.70) |
| | CAD | 13.67 (±1.18) | 3.51 (±0.49) | 15.10 (±1.22) | 5.48 (±0.44) | 1.14 (±0.29) | 6.16 (±0.50) | **15.99** (±0.76) | **3.46** (±0.34) | **21.85** (±0.87) |
| | TaBoo | **17.13** (±1.34) | **3.89** (±0.47) | **19.00** (±1.42) | **11.33** (±0.60) | **2.63** (±0.35) | **12.16** (±0.66) | 15.05 (±0.69) | 3.44 (±0.31) | 21.41 (±0.82) |

### F.6 ILLUSTRATIVE EXAMPLE OF LONG-CONTEXT TOKEN BOOSTING ON NARRATIVEQA

Figure 35 shows some examples from the NARRATIVEQA dataset to illustrate how our boosting method (Taboo) operates.

The boosted output generated by Taboo matches the ground truth, with tokens highlighted by context origin: orange tokens correspond to short-context (names from the question), while blue tokens correspond to long-context (the reasoning required from the story). The figure also shows which tokens were boosted, showing that the algorithm consistently promotes the correct words needed to form the right answer. This example illustrates how Taboo leverages long-context signals to reinforce accurate completions.

Context:
With supplies of petroleum nearly exhausted in the near future...Wez manages to board the truck and attack Max, but a head-on collision with Humungus's car kills both Wez and Humungus. Max loses control of the tanker and it rolls off the road. As the injured Max carries the Feral Kid from the wrecked tanker, he sees not petrol, but sand, leaking from the tank. The truck and its trailer are thus exposed as a decoy, allowing the other settlers to escape with the precious fuel in oil drums inside their vehicles. With Pappagallo dead, the Gyro Captain succeeds him as their chief and leads the settlers to the coast, where they establish the "Great Northern Tribe". Max remains alone in the desert, once again becoming a drifter. Years later, the Feral Kid, now the Northern Tribe's new leader (voice by Harold Baigent), reminisces about the legend of the mythical "Road Warrior" (Max) who now exists only in distant memory.

Question:
What ends up killing wez and humungus?

Ground Truth Answer:
Wez and Humungus die in a head on collison with Humungus's car

Taboo Result:
a head-on collision with Humungus's car kills both Wez and Humungus.

```
'a'    :        boost tokens = ['a', 'A'],
'head'  :      boost tokens = ['head'],
'-on'  :        boost tokens = ['-on'],
'collision'  :  boost tokens = ['collision'],
'with'  :      boost tokens = ['with'],
'Hum'  :        boost tokens = ['hum', 'Hum']
's'    :        boost tokens = ['"s', ' s']
'car'  :        boost tokens = ['car'],
'kills'  :      boost tokens = ['.', '', 'kills', '<|endoftext|>']
'both'  :      boost tokens = ['both'],
'We'    :      boost tokens = ['with'],
'.'    :        boost tokens = ['.', '<|endoftext|>']
```

Context:
Samantha Caine (Geena Davis) is a schoolteacher in the small town of Honesdale, Pennsylvania, with her boyfriend Hal (Tom Amandes) and her daughter Caitlin (Yvonne Zima). Eight years earlier, she was found washed ashore on a New Jersey beach, pregnant with Caitlin and totally amnesiac. Having never remembered her real name or any part of her life from before that day, "Samantha" has hired a number of private investigators to try to discover her past, the latest being Mitch Henessey (Samuel L. Jackson) ... In the epilogue, Charly has returned to her assumed identity of Samantha Caine, moving with Caitlin and Hal to a remote farmhouse with a bunch of goats, and declines an offer from the president to rejoin the CIA. For his part, Mitch enjoys the publicity attracted by his role in the crisis, and is last seen being interviewed by Larry King on television, where they discuss Perkins, who was indicted for treason.

Question:
Where was samantha washed ashore eight years earlier?

Ground Truth Answer:
New Jersey

Taboo Result:
"Samantha" was found washed ashore on a New Jersey beach, eight years earlier.

```
'"'    :        boost tokens = ['The', 'Sam'],
'Sam'  :        boost tokens = ['Sam'],
'"'    :        boost tokens = ['Ca', '"'].
'was'  :        boost tokens = ['(', 'was'],
'found'  :      boost tokens = ['was', 'found', 'hes'],
'was'  :        boost tokens = ['was'],
'on'   :        boost tokens = ['on'],
'a'    :        boost tokens = ['a'],
'New'  :        boost tokens = ['New'],
'Jersey'  :    boost tokens = ['Jersey'],
'.'    :        boost tokens = ['.', ','],
'eight'  :      boost tokens = ['pre', 'near'],
'earlier'  :    boost tokens = ['prior']
```

Figure 35: Example from NarrativeQA with Taboo. Orange tokens are short-context (names from the question), blue tokens are long-context (story-based reasoning). Boosted tokens align with the correct answer, showing that Taboo consistently promotes the right completions.

