# OpenReview forum: "Short-Context dominance: How Much Local Context  Natural Language Actually Needs?"
_ICLR.cc/2026/Conference — ICLR 2026 Conference Desk Rejected Submission_

### Official Review · Reviewer_fk5D · 2025-10-17

**Soundness:** 3
**Presentation:** 4
**Contribution:** 3
**Rating:** 6
**Confidence:** 3

**Summary:**

The paper highlights an important phenomenon for the NLP community.
That is, the next token prediction in popular language models (such as Llama 3, Qwen 2, Mistral) mostly relies on local context (<= 100 tokens).
The authors quantify this on popular datasets, such as GovReport, News, Wikipedia, and across 8 languages (including English, Arabic, French, German, Chinese --- Figure 11 in Appendix C.2).
Additionally, the authors propose a JSD-based method for detecting tokens with long context dependencies and Taboo, a method for accelerating long context performance. What is important to note here is that none of the methods requires the knowledge of the next token. Instead, they rely on the measure of difference between $p( . |t_1, ..., t_n)$ and $p(.| t_{n-p} ... {t_n})$.

The phenomenon is somewhat expected. However, this should not downweight the work's importance, as proper quantification is important.

**Strengths:**

+ Quantifies an important phenomenon for the NLP field
+ Provides effective methods for identifying and exploiting long context dependencies, which can significantly aid the creation of pre-training data.
+ Shows compelling evidence and provides a useful mathematical framework for analysis of long/short context dependence.

**Weaknesses:**

The work would benefit from:
- quantification of how much of the short context dominance comes from word breaking by the tokenizer
- per word type quantification of short context dominance, for example, for verbs, adjectives, nouns
-  additional quantification of the phenomenon on code

**Questions:**

Can authors quantify how much of the short context dominance comes from the word breaking by the tokenizer? What are the types of words that showcase strong short context dominance?

---

> ### Author Response · Authors · 2025-11-21
>
> Thank you for your  positive feedback and your insightful questions! We found the raised points valuable and have carefully examined them. Our clarifications and supporting results are provided below.
>
> 1- **Impact of tokenization**: This is indeed an interesting point To study it, we analyze short-context dominance separately across three token categories (Section C.3, Fig. 15): **full-word tokens**, **sub-word tokens** that **start a word**, and **sub-word tokens** that **appear in the middle or end of a word**. Our results show no noticeable differences in their MCL distributions, and all categories exhibit a similar degree of short-context dominance. Since the effect persists even when considering only full-word tokens, we conclude that short-context dominance is not an artifact of tokenization or word segmentation.
>
> 2- **Part of Speech analysis**: This is another interesting question. Since our MCL analysis is token-based rather than word-based, we map each relevant token to one of the POS categories—Noun, Verb, Adjective, or Adverb—based on the word it belongs to. The idea is that if different POS categories exhibit distinct MCL behavior, this should be reflected in the tokens associated with those words. As shown in the newly added Section C.4 (Fig. 16) of the appendix, short-context dominance appears consistently across all POS categories. However, tokens associated with adjectives and adverbs tend to require slightly longer context.
>
> 3- **Short-context dominance for code datasets**: Thank you for this recommendation. We added results on the LCC_Python dataset in Fig. 14 of Section C.2 in the appendix, focusing on longer contexts. The MCL distribution consistently follows the short-context dominance trend. Interestingly, compared to some of our natural-language long-context datasets (such as BookSum and GovReport), we find that code exhibits a noticeably higher degree of long-context reliance from an MCL perspective.
>
>
>
> 4- **Patterns regarding individual tokens**: We attempted to analyze this by tracking, for each token, whether it tends to be short-context (MCL < 64) or long-context conditioned on its preceding context. Concretely, for each token we compute $ \frac{\text{ num times the token is long-context}}{\text{ num times the token appears in the MCL analysis}} $ . We conducted this experiment on 1,000,000 context/next-token pairs from the Reddit Writing Prompts dataset using Qwen-2-7B. Below we show examples of tokens most frequently appearing as long- or short-context dependent:
>
> **High-frequency long-context tokens:** handwritten, print, History, Early, engineers, Roma, grand, sulfate, codes, black, Pre, aspir, _w
>
> **High-frequency short-context tokens:** -states, ished, Comm, axis, ological, ruler, prefix, without, itarian, easy, differently, concerned
>
> We also examined word-cloud visualizations, but did not observe a clear semantic or structural pattern distinguishing the two groups. This aligns with the lack of strong patterns in our sub-word token and POS analyses, suggesting that long- vs. short-context behavior is primarily a property of the *context sequence* and less of the actual ground truth next token. We believe such observations further encourage the focus on the next token distribution in contexts, motivating our ideas around DaMCL and LSDS.

---

> > ### Comment · Reviewer_fk5D · 2025-11-21
> > **Weaknesses addressed, questions answered**
> >
> > Thank you for the response. Considering the additional evaluation, I increase my score to 8 and strongly recommend acceptance of this paper.

---

> ### Author Response · Authors · 2025-11-22
>
> Thank you for your quick response and for the encouraging comments !

---

### Official Review · Reviewer_b2nx · 2025-10-19

**Soundness:** 2
**Presentation:** 2
**Contribution:** 2
**Rating:** 4
**Confidence:** 2

**Summary:**

In this paper, the authors tackle the problem of measuring natural-language local-context dependency and introduce Minimal Context Length (MCL), which quantifies how much local context an LLM needs to confidently predict the ground-truth next token. Building upon this, they further propose Distributionally-Aware Minimal Context Length (DaMCL), which removes the need for ground-truth labels. Together, these two measurements demonstrate that, for most sequences, a small local prefix suffices to predict their next tokens.

Motivated by this observation, the authors introduce Targeted Boosting (TaBoo), an intuitive decoding algorithm that counters short-context bias by identifying and boosting tokens that are long-context-relevant. Across the Llama, Mistral, and Qwen families, TaBoo consistently improves performance on multiple QA tasks.

**Strengths:**

1. Authors introduce the MCL, DaMCL, and LSDS metrics, which offer valuable insight into how LLMs process long contexts.
2. They further propose the enhanced decoding strategy, TaBoo, achieving consistent gains on NarrativeQA, HotpotQA, and MultifieldQA.

**Weaknesses:**

1. The experiments are conducted on outdated checkpoints. Please update them with recent models such as Llama-3.1-8B, Llama-3.2-3B, Mistral-v0.3-7B, or Qwen3-8B. Moreover, the evaluation setting is rather old-fashioned. A thorough evaluation on more comprehensive and challenging datasets is better. If evaluation is required with rich long-range dependencies, LEval [1] or LongBench-v2 [2] should be preferred. Besides, including the more challenging RULER benchmark [3] would also help verify whether TaBoo remains robust when the context contains misleading information.
2. While I find the definitions and observations of MCL and DaMCL valuable, the extensive analysis leaves little space for showing how these insights can be translated into practical improvements. I also struggle to see why the metrics are used for an enhanced decoding strategy. This may stem from my limited familiarity with contrastive decoding. I would appreciate a clearer explanation.
3. Figure 7 is blurry. Please check and replace it with a higher-resolution version.

[1] L-Eval: Instituting Standardized Evaluation for Long Context Language Models https://arxiv.org/abs/2307.11088

[2] LongBench v2: Towards Deeper Understanding and Reasoning on Realistic Long-context Multitasks https://arxiv.org/abs/2412.15204

[3] RULER: What's the Real Context Size of Your Long-Context Language Models? https://arxiv.org/abs/2404.06654

**Questions:**

See Weaknesses

---

> ### Author Response · Authors · 2025-11-21
>
> We appreciate your feedback and are glad that you find our work’s insights into understanding LLM behavior valuable. Below, we address the points raised in your review.
>
> 1- Prompted by your question, we have now included experimental results on additional models and datasets. We used Mistral-v0.3-7B and base Qwen-3-8B to run the MCL experiments on our updated long-context datasets, and we observe results consistent with our earlier findings (see Fig. 35 in Appendix F.6). For the TaBoo experiments, we appreciate the recommendation to use L-Eval , as it includes several relevant long-context datasets we can use! We evaluated on the NaturalQA, MultiDocQA, and ScientificQA subsets. We haven’t used RULER (as it is primarily synthetic) or LongBench-v2 (as it is multiple-choice QA and less compatible with our pipeline). The additional results are shown in Table 10, where we find that—even though the dataset is more challenging due to its complexity and longer contexts—TaBoo still improves performance compared to vanilla decoding and CAD.
>
> 2- Our goal in this work is to characterize the short-context bias present in natural language and in LLMs, initially through MCL, and then through distribution-based measures such as DaMCL and LSDS. A key advantage of DaMCL/LSDS is that they allow determining the short- vs long-context nature of a sequence **without requiring the ground-truth next token**, enabling analysis across every position in a sequence.
>
> TaBoo is such an example. From the MCL and DaMCL analysis, we observe a consistent short-context dominance that can negatively affect long-context performance. LSDS further allows us to detect which tokens truly rely on long-context information. TaBoo leverages this by identifying tokens whose probabilities shift when moving from the short-context to the full-context NTP distributions and selectively up-weighting these long-context-relevant tokens during inference.
>
> While TaBoo is a contrastive decoding method, its motivation and mechanics differ from CAD. CAD contrasts **no-context vs full-context** distributions to mitigate hallucinations. In contrast, TaBoo specifically targets the **short- vs long-context** bias revealed by our analysis, compares **short-context vs full-context** distributions, and applies adjustments **only for sequences identified as long-context dependent**, while selectively modifying tokens that exhibit long-context relevance. We also include ablations in **Section F** that support these design choices. We provide a more in-depth discussion of contrastive decoding and how TaBoo differs from prior approaches in **Appendix Section B**.
>
> More broadly, our analysis motivates additional techniques for mitigating short-context bias or exploiting short-context structure. As suggested by Reviewer SBU7 and mentioned in our Conclusions, one promising direction is training-time interventions—for example, emphasizing long-context sequences during pre-training to improve long-context behavior. Another direction involves using the knowledge of short-context dominance to accelerate inference by leveraging the fact that most next-token predictions rely primarily on local context (for example, by integrating with speculative decoding methods).
>
> 3- Thank you for catching that! . We have updated the blurry image.
>
>
> We hope that these comments addressed your concerns. If you have remaining questions, we would be happy to respond. Thanks again for your time.

---

> > ### Comment · Reviewer_b2nx · 2025-11-22
> >
> > Thank you for adding the experiments and clarifications. The added experiments demonstrate that your method remains effective on newer models. I hope the authors can update this in the main text. Furthermore, I hope the authors can report scores of comprehensive benchmarks as much as possible, instead of specific tasks. These will help demonstrate the effectiveness and value of our work to a wider audience. Considering the authors' sincere response, I have increased the paper's score to 6.

---

> > > ### Author Response · Authors · 2025-11-23
> > >
> > > Thank you for your encouraging response and increasing the score. We will make sure to include the new models, datasets and additional discussion in the camera ready version.

---

### Official Review · Reviewer_jngf · 2025-10-25

**Soundness:** 4
**Presentation:** 3
**Contribution:** 3
**Rating:** 8
**Confidence:** 4

**Summary:**

This paper investigates the short-context dominance hypothesis in Large Language Models (LLM), which proposes that for the majority of sequences, a small local context suffices to predict the next token accurately. Through extensive experiments, the authors find that about 75–80% can be predicted with no more than the last 96 tokens. The paper introduces two key concepts: Minimal Context Length (MCL) and Distributionally Aware MCL (DaMCL), the latter of which does not require ground-truth knowledge of the next token. These methods allow for the detection of short and long-context sequences. Moreover, the paper introduces a contrastive-decoding-like decoding algorithm, TaBoo, which addresses the bias induced by short-context dominance in LLMs, boosting performance in tasks with long-context dependencies such as question answering.

**Strengths:**

1. **Insightful Findings** The authors proposed the short-context dominance hypothesis, and backed the statement by substantial experimental evidence. On a series of LLMs, most sequences rely on very localized context, even in long documents, thus challenging the idea that large context windows are always necessary for accurate token prediction.
2. **Applicability.** The introduction of MCL and DaMCL provides a new approach to understanding and measuring context dependency in LLMs. DaMCL, in particular, offers a practical method for detecting long-context dependency without ground-truth knowledge. The proposed metrics are highly applicable, working across different models and various datasets, including both short and long-context tasks
3. **Promising performance gains for TaBoo.** TaBoo effectively boosts the performance of LLMs in downstream tasks, offering a more delicate solution to handling context compared to existing contrastive decoding methods.

**Weaknesses:**

1. **Efficiency Analysis.** Efficiency of the proposed method could be further analyzed and reported. It would be better if the authors could report the efficiency (in seconds) before/after applying TaBoo to further demonstrate the extra computational gains required by TaBoo.
2. (Optional, as the authors left this for future works in L169.) **Performance on Reasoning Tasks.** I am curious about the distribution of short/long-context dependency, and the performance of TaBoo on more difficult reasoning tasks, such as Math and Code. This is completely optional and does not affect my overall rating to this article.

**Questions:**

See weaknesses.

---

> ### Author Response · Authors · 2025-11-21
>
> We are glad you found the paper interesting, and we are particularly encouraged by your positive assessment. Below, we address and clarify the main points you raised in your review.
>
>
> **Efficiency Analysis**: The primary source of overhead in TaBoo comes from the second forward pass required to compute LSDS (Step 1 in Algorithm 1). In Fig. 32, we compare the runtime of a single full-context forward pass with that of two passes (full context + short sub-context). The difference is marginal, especially for longer contexts. This is expected given (i) active KV-caching during inference and (ii) the short length of the sub-context (32–64 tokens), which makes the additional pass comparatively inexpensive. To highlight this more explicitly, we have added Fig. 33 in Appendix Section E.4 which reports the cost of LSDS computation and long-context relevant token detection; both operate on the order of milliseconds and are therefore negligible relative to the overall inference time.
>
> **Explorations on reasoning**: This is a very nice idea; thank you! Applying TaBoo in this setting would require running the algorithm during CoT generation itself, actively promoting long-context–relevant tokens throughout the reasoning process. This is beyond our current implementation, which focuses on Q/A and short-summary generation, and would involve evaluating TaBoo in the context of full long-form generation. Before extending TaBoo to improve CoT, following our methodology, we would first aim to better understand long-text generation and how to quantitatively evaluate improvements that arise from promoting long-context tokens throughout generation.
>
> As a very preliminary step, we analyze the short-context dominance of sequences generated in the s1: Simple Test-Time Scaling dataset (Muennighoff et al., 2025 [1]) using Qwen-2.5-7B. As shown in the newly added Fig. 11 of Appendix C.2, the MCL distribution on these generated CoT sequences indicates that short-context dominance persists in this reasoning setting. Combined with our results on MultiFieldQA (a multi-hop reasoning benchmark) in Table 1, these findings indeed motivate further investigation into how methods like TaBoo may benefit reasoning tasks. We will include this as a promising future direction in the conclusion.
>
> [1] s1: Simple Test-Time Scaling — https://arxiv.org/abs/2501.1939

---

> > ### Comment · Reviewer_jngf · 2025-11-24
> >
> > Many thanks to the authors for their prompt response. My concerns have been successfully resolved.

---

### Official Review · Reviewer_SBU7 · 2025-11-01

**Soundness:** 3
**Presentation:** 4
**Contribution:** 4
**Rating:** 8
**Confidence:** 4

**Summary:**

This paper presents the “short-context dominance” hypothesis, that most natural-language predictions depend mainly on local context rather than long-context information. Using the LLM as a statistical oracle, the authors measure each sequence’s Minimal Context Length (MCL) and find that about 80% of next-token predictions require fewer than 96 preceding tokens. To address the stochastic nature of the practical generation process, they extend this with a distribution-aware variant (DaMCL). In addition, the paper proposes a Long-Short Distribution Shift (LSDS) that identifies long-context sequences using a comparative approach (local vs. global). Building on these insights, the paper introduces TaBoo (targeted boosting), a decoding algorithm that selectively boosts tokens needing long-range context. Experimental results demonstrate that TaBoo consistently outperforms baseline decoding methods.

**Strengths:**

* The paper clearly illustrates that the key challenge in long-context modeling is not merely handling long text inputs, but distinguishing cases that truly require long-range contextual understanding. This insight has been relatively underexplored and is an important conceptual contribution.
* The Related Work section, particularly the discussion on N-gram language models, is comprehensive and well-situated within prior research.
* The logical progression from MCL to DaMCL, LSDS, and finally to token-level boosting (TaBoo) is coherent and intuitive. The paper is engaging and easy to follow.

**Weaknesses:**

* (minor) Because decoder-only LLMs predict the next token at every position, the analysis based on MCL and DaMCL, which evaluates prediction accuracy at selected tokens, may not fully capture the dynamic behavior of models over entire documents.

**Questions:**

* Both MCL and DaMCL depend on the choice of the LLM as an oracle. Although the authors partially address model robustness through cross-model comparisons, a discussion on how to mitigate dependence on a specific LLM would strengthen the paper.
* Rather than applying token-level boosting, one might consider increasing the proportion of long-context-dependent samples in the training data. Could such a data-level intervention provide similar benefits?
* (minor) There are some inconsistencies in citation formatting (e.g., Section 4.1).

**Details Of Ethics Concerns:**

No concerns.

---

> ### Author Response · Authors · 2025-11-21
>
> We are happy that you found the paper engaging and appreciate the positive feedback. Below, we address the points raised in your review.
>
> **Choice of LLM**: This is an excellent point. As you rightly point out, we already present all our results for a suite of models towards ensuring “robustness” of the findings. Following your prompt, to further assess how sensitive our long-context classification is to the choice of oracle model, we now measure cross-model agreement on LSDS-based long/short **context** decisions. For this, we sample 10K contexts from Wikipedia and classify each context as long- or short-context using five different LLMs: Llama-3-8B, Mistral-v0.3-7B, OLMo-2-7B, Qwen-2.5-7B, and InternLM-2.5-7B (all of similar size ), using a threshold of $\tau = 0.5$.
>
> Across all 10 pairwise comparisons, the models agree on roughly **80–86%** of classifications:
>
> llama3 vs qwen2: 83.00
> llama3 vs mistral: 82.67
> llama3 vs olmo2: 81.71
> llama3 vs internlm: 82.60
> qwen2 vs mistral: 85.82
> qwen2 vs olmo2: 85.44
> qwen2 vs internlm: 85.49
> mistral vs olmo2: 84.51
> mistral vs internlm: 85.98
> olmo2 vs internlm: 83.93
>
> Using a majority vote across the five models as a reference, we find that in **87%** of contexts, at most one model disagrees with the majority label, and in **66%** of contexts all models unanimously agree. While some variation exists—likely due to differences in tokenization, pretraining data, or model architecture—the overall level of agreement suggests that the long/short context classification is not strongly tied to any particular LLM and reflects a consistent signal across different models.
>
>
> **Training with long-context**: We very much agree and believe this is an insightful direction; in fact, one that we have been considering ourselves. Using our metrics to guide training—for example, by over-sampling contexts with larger MCL/DaMCL values or up-weighting their loss—could help counteract the short-context bias of natural language that implicitly affects long-range performance. We also note that existing data-curation strategies that emphasize long documents are likely selecting samples with higher MCL/DaMCL, and as we mentioned in the paper conclusion, plan to study this connection explicitly in future work.
>
> **Citation Format**: We apologize for the formatting issue and appreciate the reviewer pointing it out. The revised version now includes the corrected citation formatting.

---

> > ### Comment · Reviewer_SBU7 · 2025-11-25
> >
> > Thank you for the response and additional experiments.
> > As the score is already high, I'd maintain my evaluation.

---

> > > ### Author Response · Authors · 2025-11-25
> > >
> > > We appreciate your thoughtful input and thank you for your encouraging review.

---

### Note · Program_Chairs · 2026-01-17
**Submission Desk Rejected by Program Chairs**

The following references in this submission do not refer to real documents and/or have major errors in bibliographic information:

 Bram van der Poel, Zheng Li, Shashi Narayan, Ramesh Nallapati, Kathleen McKeown, and Trung Bui. Mutual information decoding for abstractive summarization. In Proceedings of the 2022 Conference on Empirical Methods in Natural Language Processing, pp. 5850-5866. Association for Computational Linguistics, 2022.